# LRRC23 truncation impairs radial spoke 3 head assembly and sperm motility underlying male infertility

Jae Yeon Hwang[1,2†], Pengxin Chai[3†], Shoaib Nawaz[4†‡], Jungmin Choi[5,6], Francesc Lopez-Giraldez[7], Shabir Hussain[8§], Kaya Bilguvar[5,7], Shrikant Mane[6], Richard P Lifton[9], Wasim Ahmad[4,8], Kai Zhang[3], Jean-Ju Chung[1,10]*

[1]Department of Cellular and Molecular Physiology, Yale School of Medicine, Yale University, New Haven, United States; [2]Department of Molecular Biology, Pusan National University, Busan, Republic of Korea; [3]Department of Molecular Biophysics and Biochemistry, Yale School of Medicine, Yale University, New Haven, United States; [4]Department of Biotechnology, Faculty of BiologicalSciences, Quaid-i-Azam University, Islamabad, Pakistan; [5]Department of Genetics, YaleSchool of Medicine, Yale University, New Haven, United States; [6]Department of Biomedical Sciences, Korea University College of Medicine, Seoul, Republic of Korea; [7]Yale Center forGenome Analysis, Yale University, West Haven, United States; [8]Department of Biochemistry, Faculty of Biological Sciences, Quaid-i-Azam University, Islamabad, Pakistan; [9]Laboratory of Human Genetics and Genomics, The Rockefeller University, New York, United States; [10]Department of Obstetrics, Gynecology and Reproductive Sciences, Yale School of Medicine, Yale University, New Haven, United States

*For correspondence: jean-ju.chung@yale.edu

[†]These authors contributed equally to this work

Present address: [‡]Department of Human Genetics, Sidra Medicine, Doha, Qatar; [§]Clinical and Molecular Metabolism Program, Faculty of Medicine, University of Helsinki, Helsinki, Finland

**Abstract** Radial spokes (RS) are T-shaped multiprotein complexes on the axonemal microtubules. Repeated RS1, RS2, and RS3 couple the central pair to modulate ciliary and flagellar motility. Despite the cell type specificity of RS3 substructures, their molecular components remain largely unknown. Here, we report that a leucine-rich repeat-containing protein, LRRC23, is an RS3 head component essential for its head assembly and flagellar motility in mammalian spermatozoa. From infertile male patients with defective sperm motility, we identified a splice site variant of *LRRC23*. A mutant mouse model mimicking this variant produces a truncated LRRC23 at the C-terminus that fails to localize to the sperm tail, causing male infertility due to defective sperm motility. LRRC23 was previously proposed to be an ortholog of the RS stalk protein RSP15. However, we found that purified recombinant LRRC23 interacts with an RS head protein RSPH9, which is abolished by the C-terminal truncation. Evolutionary and structural comparison also shows that LRRC34, not LRRC23, is the RSP15 ortholog. Cryo-electron tomography clearly revealed that the absence of the RS3 head and the sperm-specific RS2-RS3 bridge structure in LRRC23 mutant spermatozoa. Our study provides new insights into the structure and function of RS3 in mammalian spermatozoa and the molecular pathogenicity of LRRC23 underlying reduced sperm motility in infertile human males.

## eLife assessment

This study provides **valuable** findings on a causative relationship between LRRC23 mutations and male infertility due to asthenozoospermia. The evidence supporting the conclusions is **solid**. This work will be of interest to biomedical researchers who work on sperm biology and non-hormonal male contraceptive development.

## Introduction

Motile cilia and flagella are evolutionarily conserved organelles essential for cellular motility (*Inaba, 2011*; *Ishikawa, 2017*). The core of motile cilia and flagella is the '9+2' axoneme, characterized by a scaffold structure composed of nine peripheral microtubule doublets (MTDs) and a central pair of singlet microtubules (CP). Each MTD binds two rows of dyneins, the outer-arm dyneins (OAD) and inner-arm dyneins (IAD), which generate mechanical force required for the ciliary beating via ATP hydrolysis (*Kubo et al., 2021*; *Rao et al., 2021*). Radial spoke (RS) controls the amplitude of the ciliary and flagellar beat by transmitting mechanochemical signals from the CP to the axonemal dyneins (*Smith and Yang, 2004*; *Viswanadha et al., 2017*). In *Chlamydomonas reinhardtii*, RS mutations paralyze the flagellar movement (*Witman et al., 1978*) and the axoneme lacking RS system shows reduced velocity of microtubule sliding by dyneins (*Smith and Sale, 1992*). In human and mouse, mutations in RS components or their absence cause primary ciliary dyskinesia (PCD) and/or male infertility due to the defective ciliary and flagellar beating (*Abbasi et al., 2018*; *Liu et al., 2021*; *Sironen et al., 2020*). These studies highlight the physiological importance of RS in regulating ciliary and flagellar movement.

A triplet of three RSs (RS1, RS2, and RS3) repeats every 96 nm along the peripheral MTDs and this 96 nm periodicity of RS is well-conserved in cilia and flagella across diverse organisms (*Viswanadha et al., 2017*; *Zhu et al., 2017*). RS is a multiprotein complex; at least 23 RS components were identified in *C. reinhardtii* (*Gui et al., 2021*; *Viswanadha et al., 2017*; *Yang et al., 2006*). These RS components are also crucial for RS organization and function in other ciliated and flagellated organisms (*Bazan et al., 2021*; *Ralston et al., 2006*; *Sironen et al., 2020*; *Urbanska et al., 2015*), indicating evolutionarily conserved molecular composition of the RS. However, cryo-electron microscopy (cryo-EM) studies have revealed species- and tissue-specific structural variabilities in RS3. For example, RS3 in most species is a T-shaped axonemal structure that consists of a head and a stalk like those of RS1 and RS2 (*Imhof et al., 2019*; *Leung et al., 2021*; *Lin et al., 2014*; *Pigino et al., 2011*) but RS3 in *C. reinhardtii* is a headless and stump-like structure (*Pigino et al., 2011*). In addition, mammalian sperm flagellar axoneme carries a unique bridge structure between RS2 and RS3 (*Leung et al., 2021*), which is not observed from tracheal cilia as well as other flagellated organisms (*Imhof et al., 2019*; *Lin et al., 2014*). This variability suggests that RS3 is an evolutionarily more divergent structure and conveys species- and/or tissue-specific ciliary and flagellar function. Yet, the overall molecular composition of RS3 and the extent to which RS3 components are species- and/or tissue specific remains largely unknown.

Asthenozoospermia (ASZ) is the male infertility classified by reduced sperm motility (*World Health Organization, 2010*). Approximately 80% of male infertile patients manifest sperm motility defects (*Curi et al., 2003*) as infertile male with abnormal sperm morphology and/or reduced sperm count often accompany low motility (*Cavarocchi et al., 2022*; *Touré et al., 2021*). Recent whole exome sequencing studies identified genetic defects in RS components from idiopathic ASZ patients (*Liu et al., 2021*; *Martinez et al., 2020*; *Shen et al., 2021*). Mutant analyses using model organisms further elucidated the function of RS components in the flagellar movement and the pathogenic mechanisms (*Liu et al., 2021*; *Martinez et al., 2020*). Thus, WES combined with functional and structural analyses of mutants, especially in a mouse model, would be a powerful and direct approach to understand mammalian specific RS3 roles including sperm motility regulation and genetic etiologies causing male infertility.

Here, we report a bi-allelic loss-of-function splicing variant in *LRRC23* from ASZ patients in a Pakistani consanguineous family. We generated a mouse model that mimics the human mutation and found that the mutation leads to C-terminal truncated LRRC23 and that the mutant mice phenocopy the impaired sperm motility and male infertility. Using biochemical analyses and structural prediction, we showed that, different from previously known, LRRC23 is not the ortholog of a RS2 stalk protein RSP15 but interacts with a known RS head protein. Finally, we visualized the in-cell structures of RS triplets in intact WT and mutant sperm using cryo-electron tomography (cryo-ET). We observed missing RS3 head and aberrant junction between RS2 and RS3 in the mutant flagellar axoneme, unarguably demonstrating that LRRC23 is a head component of RS3. This study provides molecular pathogenicity of *LRRC23* in RS-associated ASZ and reveals unprecedented structural insights into the RS3 and its implication in normal sperm motility and male fertility in mammals.

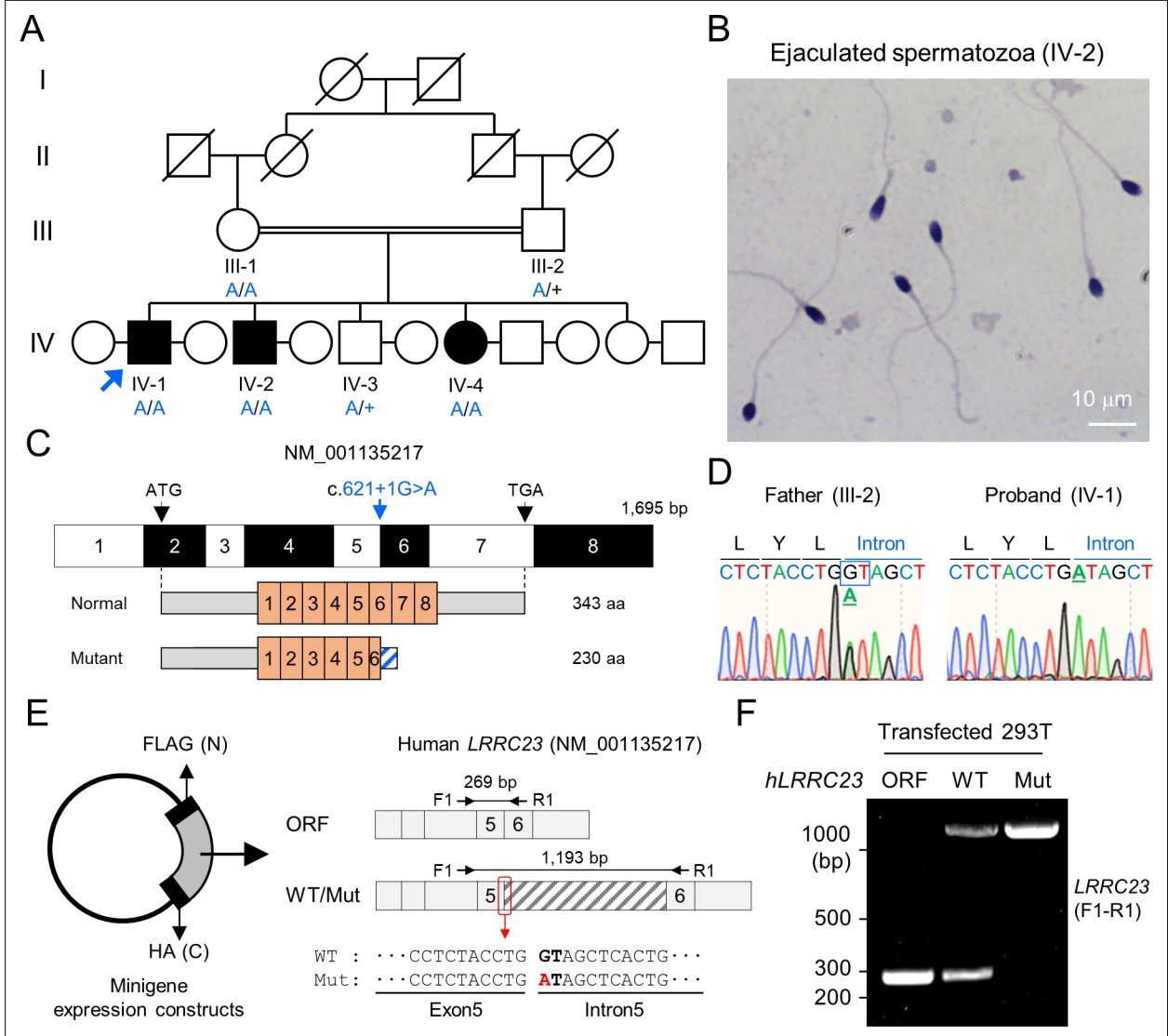

**Figure 1.** A bi-allelic splicing donor site variant in *LRRC23* was identified from asthenozoospermia patients. (**A**) A consanguineous pedigree with two infertile males (IV-1 and IV-2). IV-1 was subjected for WES (arrow). Genotypes of the variant (blue) in all family members included in this study (III-1, III-2, IV-1, IV-2, IV-3, and IV-4) are confirmed by Sanger sequencing. +, wild-type allele. An infertile female sibling (IV-4) is marked in black circle. (**B**) Papanicolaou-stained sperm from the infertile male (IV-2). (**C**) Mapping of the LRRC23 variant. Mutation of G to A at the splicing donor site in the 5th intron is predicted to prevent *LRRC23* mRNA from splicing. (**D**) Sequencing chromatograms presenting the *LRRC23* variant in the infertile male (IV-1) and his father (III-2). The variant is underlined and normal splicing donor site (GT) is boxed. (**E, F**) Minigene assay for testing altered splicing of *LRRC23* by the variant. (**E**) Minigene constructs expressing *LRRC23* ORF containing the 5th intron (sashed) with wild-type (WT) or mutant (Mut, red) splicing donor site were generated. The constructs are tagged with FLAG and HA at N- and C-termini, respectively. (**F**) RT-PCR of the 293T cells transfected with the minigene constructs reveals the 5th intron is not spliced out and retained by the variant. Intron-spanning primers, F1 and R1, are used. Repeated three times with biological replications.

The online version of this article includes the following source data and figure supplement(s) for figure 1:

**Figure supplement 1.** A *LRRC23* splicing site variant identified from infertile male patients.

**Figure supplement 1—source data 1.** Uncropped blot images for *Figure 1—figure supplement 1*.

## Results

### A loss-of-function splice site variant truncating LRRC23 is identified from a consanguineous family with asthenozoospermia

A consanguineous Pakistani family with infertile males was recruited (*Figure 1A*). Both infertile males (IV-1 and IV-2) have failed to have child over 3 years of unprotected sex after marriage (***Supplementary***

*file 1*). The male infertile patients do not show PCD-related symptoms, abnormal heights, weights, secondary characteristics, nor anatomical defects. They also have normal karyotypes without Y chromosome microdeletion. Overall, the Papanicolaou (PAP) stained spermatozoa from the infertile patients showed an overall normal morphology (*Figure 1B* and *Figure 1—figure supplement 1*), which was further supported by normal ranges of abnormal sperm morphology assessed by clinical semen analysis (*Supplementary file 1*). However, their progressive motility was lower than the World Health Organization (WHO) standard (*World Health Organization, 2010*) and the patients were clinically diagnosed as ASZ (*Supplementary file 1*). To understand the genetic etiology underlying the defective sperm motility, we performed whole exome sequencing (WES) on the proband, IV-1. WES estimated 5.02% inbreeding co-efficiency and the longest homozygous-by-descent segment as 37.5 cM, verifying the consanguineous nature of the recruited family. Among the four identified rare variants (*Supplementary file 2*), only one homozygous splicing donor site variant (c.621+1 G>A) in *LRRC23* (leucin-rich repeat containing protein 23) is co-segregated with the male infertility phenotype (*Figure 1A, C and D*). Of note, a female sibling (IV-4) who also has the homozygous *LRRC23* variant (IV-4) is infertile because her partner was able to have children from his second marriage (*Figure 1A*). However, the infertility of IV-4 is not likely be due to the variant because the mother (III-1) also carries the homozygous allele but was fertile (*Figure 1A*). The variant at the splicing donor site of *LRRC23* intron 5 is predicted to prevent splicing out of the intron, which can lead to early termination of protein translation with loss of 136 amino acids at the C-terminus (*Figure 1—figure supplement 1B, C*). To verify the splicing defects and generation of mutant protein by the variant, we constructed minigenes to express *LRRC23* ORF spanning partial intron 5 with the normal or variant sequence at the splicing site in 293T cells (*Figure 1E*). 293T cells transfected with the construct carrying the variant failed to splice out the intronic region (*Figure 1F*) and generated only truncated LRRC23 (*Figure 1—figure supplement 1D*). These results suggest the variant is pathogenic and explains the male infertility with defective sperm motility in this family.

## C-terminally truncated LRRC23 fails to localize in the flagella and causes defective sperm motility

LRRC23 is a radial spoke (RS) component (*Padma et al., 2003*). Consistent with the ASZ phenotype in our infertile patients with a new splice mutation in *LRRC23*, genetic ablation of *Lrrc23* in mice causes severe sperm motility defects and male infertility (*Zhang et al., 2021*). However, how the C-terminus truncation of LRRC23 affects the RS structure and function remains unresolved. To better understand detailed function of LRRC23 and the molecular pathogenicity of the identified variant, we generated *Lrrc23* mutant mice by CRISPR/Cas9 genome editing to mimic the predicted outcome in human patients (*Figure 1—figure supplement 1* and *Figure 2—figure supplement 1*). We targeted two regions, one in intron 5 and the other in intron 7 (*Figure 2—figure supplement 1B*) to delete exon 6 and 7 together and express truncated LRRC23 at C-terminus. We established two mouse lines with 4126 or 4135 bp deletion (*Lrrc23-4126del* and *Lrrc23-4135del*, respectively; *Figure 2—figure supplement 1C*). Both homozygous *Lrrc23-4126del* and *4135del* mice displayed the identical male infertility and defective sperm motility phenotypes in our initial characterization. In this study, we used *Lrrc23-4126del* line as *Lrrc23-mutant* line unless indicated. We observed that truncated *Lrrc23* mRNA is expressed from the mutant *Lrrc23* allele but the total mRNA level of *Lrrc23* in testis is not different from wildtype (WT) to that in *Lrrc23*$^{\Delta/\Delta}$ males (*Figure 2—figure supplement 1D–F*). Sequencing the truncated *Lrrc23* mRNA revealed that the transcript would produce mutant LRRC23 containing 27 non-native amino acids translated from 3' UTR instead of 136 amino acids at the C-terminus (*Figure 2—figure supplement 1G*). Despite the comparable *Lrrc23* mRNA levels in WT and *Lrrc23*$^{\Delta/\Delta}$ testis, the truncated LRRC23 is detected only marginally in the microsome fraction of *Lrrc23*$^{\Delta/\Delta}$ testis, different from full-length LRRC23 enriched in the cytosolic fraction of WT testis (*Figure 2A* and *Figure 2—figure supplement 2A and B*). In addition, the mutant LRRC23 is not present in epididymal sperm (*Figure 2B and C*, and *Figure 2—figure supplement 2C–E*), indicating that the C-terminal region is essential for proper LRRC23 transportation to the sperm flagella.

Sperm from *Lrrc23*$^{\Delta/\Delta}$ and *Lrrc23-4135del*$^{\Delta/\Delta}$ males are morphologically normal (*Figure 2C* and *Figure 2—figure supplement 2E and F*) and the epididymal sperm counts of *Lrrc23*$^{+/\Delta}$ and *Lrrc23*$^{\Delta/\Delta}$ males are not significantly different (*Figure 2D*). Despite the normal morphology and sperm counts, *Lrrc23*$^{\Delta/\Delta}$ males are 100% infertile (*Figure 2E and F*). By contrast, *Lrrc23*$^{\Delta/\Delta}$ females are fertile, which

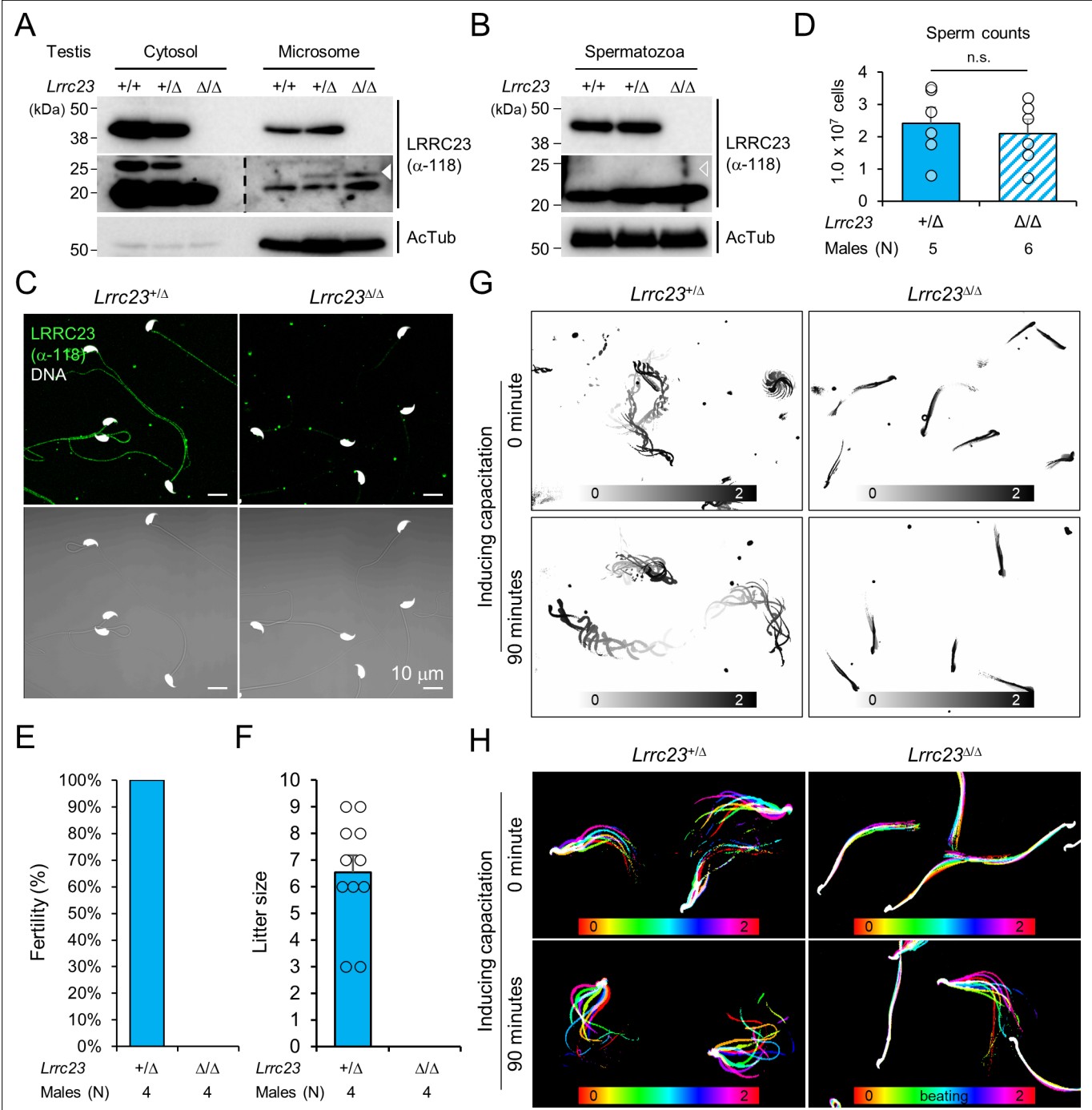

**Figure 2.** *Lrrc23* mutant mice mimicking human splice variant phenocopy male infertility and reduced sperm motility. (**A, B**) Immunoblotting of LRRC23 in testis (**A**) and epididymal sperm (**B**) from mutant male mice. Truncated LRRC23 (arrowheads) is detected from testis microsome fraction (filled), but not in mature sperm (empty), of heterozygous (+/Δ) and homozygous (Δ/Δ) males. Acetylated tubulin (AcTub) is a loading control. Experiments were performed with three biological replications. (**C**) Confocal images of immunostained LRRC23 in *Lrrc23*+/Δ and *Lrrc23*Δ/Δ epididymal sperm Experiments were repeated with three biological replications. (**D**) Epididymal sperm counts. n.s., not significant. (**E**) Pregnancy rate of *Lrrc23*+/Δ and *Lrrc23*Δ/Δ males. (**F**) Number of litters from fertile females mated with *Lrrc23*+/Δ and *Lrrc23*Δ/Δ males. (**G**) Swimming trajectory of *Lrrc23*+/Δ and *Lrrc23*Δ/Δ sperm in viscous media (0.3% methylcellulose). Swimming trajectory for 2 s is overlaid. Experiments were performed with three biological replications. See *Video 1*. (**H**) Flagellar waveforms of *Lrrc23*+/Δ and *Lrrc23*Δ/Δ sperm before (0 min) and after (90 min) inducing capacitation. Flagellar movements for two beat cycles are overlaid and color coded in time. Experiments were performed with three biological replications. See *Video 2*. In (**A–C**), samples from WT were used for positive or negative control of normal or truncated LRRC23. In (**D, F**), circles indicate sperm counts from individual males (**D**) and pup numbers from each litter (**F**), and data represented as mean ± SEM (D, Mann-whiteny U test; F, Student's t-test). n.s., non-significant.

*Figure 2 continued on next page*

*Figure 2 continued*

The online version of this article includes the following source data and figure supplement(s) for figure 2:

**Source data 1.** Uncropped blot images for *Figure 2*.

**Figure supplement 1.** Generation of LRRC23 mutant mouse models.

**Figure supplement 1—source data 1.**

**Figure supplement 2.** Characterization of the *Lrrc23* lossof-function male mice.

**Figure supplement 2—source data 1.** Uncropped blot images for *Figure 2—figure supplement 1*.

supports the IV-4's infertility is not likely due to the identified variant (*Figure 1A*). To further understand how the homozygous *Lrrc23* mutation causes male infertility, we performed Computer Assisted Semen Analysis (CASA). *Lrrc23*$^{\Delta/\Delta}$ sperm motility parameters are significantly altered compared to *Lrrc23*$^{+/\Delta}$ sperm (*Figure 2—figure supplement 2G*). *Lrrc23*$^{\Delta/\Delta}$ sperm cannot swim efficiently under viscous conditions that mimic the environment in female reproductive tract (*Figure 2G* and *Video 1*), and their flagella just vibrate but do not beat normally (*Figure 2H*, *Figure 2—figure supplement 2H* and *Video 2*). In addition, inducing capacitation did not rescue any observed motility defect of *Lrrc23*$^{\Delta/\Delta}$ sperm as demonstrated by flagellar waveform analysis, and CASA measurement of curvilinear velocity (VCL), straight line velocity (VSL), and amplitude of lateral head (ALH). These results suggest the C-terminal truncation dysregulates the flagellar localization of the mutant LRRC23, leading to sperm motility defects and male infertility.

## C-terminal truncation of LRRC23 abolishes its interaction with radial spoke head

Recent cryo-ET studies revealed that T-shaped RS structures (i.e., head and stalk) are conserved across the species (*Leung et al., 2021*; *Lin et al., 2014*; *Figure 3A*). Three RSs (RS1, RS2, and RS3) are repeated in a 96 nm interval along the flagellar axoneme in mammalian sperm, sea urchin sperm, *Trypanosoma brucei*, and even in *C. reinhardtii* with stump-like RS3 without a head structure (*Figure 3A*). LRRC23 is a RS protein in chordate axoneme and has been considered especially as the orthologue of RSP15, a RS2 stalk protein in *C. reinhardtii* (*Han et al., 2018*; *Satouh and Inaba, 2009*; *Yang et al., 2006*; *Zhang et al., 2021*; *Figure 3B*). We initially hypothesized the C-terminal truncation of LRRC23 affects the assembly of a RS stalk and/or its incorporation into the RS2. Thus, we tested the protein-protein interaction of normal (hLRRC23$^{WT}$) and mutant (hLRRC23$^{Mut}$) human LRRC23 with known RS stalk (RSPH3, RSPH22) or head (RSHP6A, RSHP9) proteins using RSPH-trap

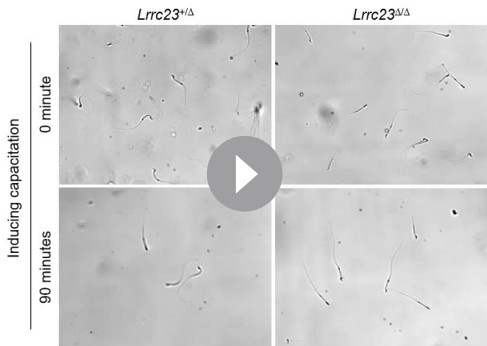

**Video 1.** *Lrrc23*$^{+/\Delta}$ and *Lrrc23*$^{\Delta/\Delta}$ sperm swimming freely in a viscous environment. Free-swimming *Lrrc23*$^{+/\Delta}$ and *Lrrc23*$^{\Delta/\Delta}$ sperm in the viscous condition containing 0.3% methylcellulose were recorded for 2 s before and after inducing capacitation for 90 min. Individual videos are played at 50 fps (1/2 speed).

https://elifesciences.org/articles/90095/figures#video1

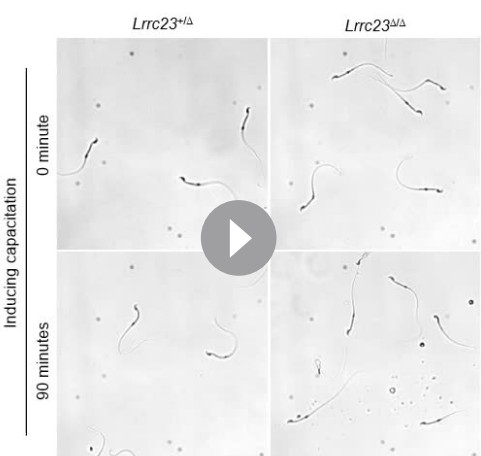

**Video 2.** Flagellar waveform of *Lrrc23*$^{+/\Delta}$ and *Lrrc23*$^{\Delta/\Delta}$ sperm before and after inducing capacitation. Tail movements of head-tethered sperm from *Lrrc23*$^{+/\Delta}$ and *Lrrc23*$^{\Delta/\Delta}$ males are recorded for 2 s before and after incubation under capacitating conditions for 90 min. Each video is played at 100 fps (1/2 speed).

https://elifesciences.org/articles/90095/figures#video2

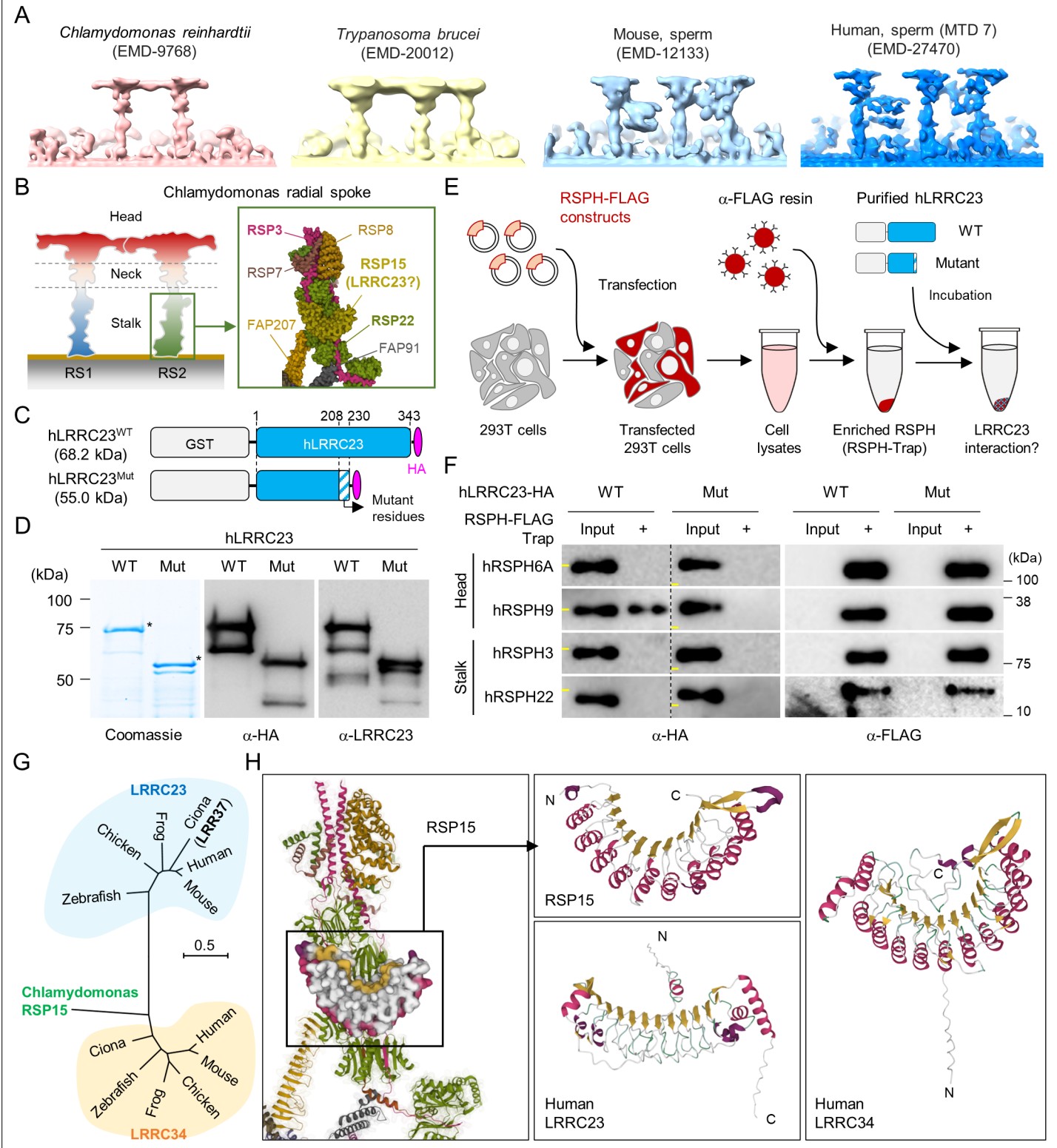

**Figure 3.** C-terminal truncation of human LRRC23 by the splicing site mutation prevents its interaction with radial spoke (RS) head. (**A**) Sub-tomogram averaging images of RSs from *Chlamydomonas reinhardtii* (*red*), *Trypanosoma brucei* (yellow), mouse sperm (sky blue), and human sperm (blue). RSs at 7th microtubule doublet (MTD) are shown for human sperm. Original data from Electron Microscopy Data Bank was rendered. (**B**) Structure of RS in *C. reinhardtii*. A schematic cartoon shows the RS1 and 2. The structure of RS2 stalk is shown in inset (PDB Id: 7JRJ). (**C, D**) Purification of normal (hLRRC23WT) and the mutant human LRRC23 (hLRRC23Mut) by the splicing site mutation (c.621+1 G>A) in this study. (**C**) Diagrams for the purified

*Figure 3 continued on next page*

Figure 3 continued

recombinant normal and mutant proteins tagged with tagged with GST and HA at N- and C-termini, respectively. (**D**) Purified proteins by Coomassie blue staining (*left*) and immunoblotting with α-HA (*middle*) and a-LRRC23 (*right*). Proteins matched to the predicted size were marked with asterisks. (**E**) A cartoon of the RSPH-trap approach to test LRRC23 interaction with RS proteins. Individual human RS proteins tagged with FLAG (RSPH-FLAG) are expressed in 293T cells and enriched by α-FLAG resin from cell lysates. The recombinant RSPH proteins were incubated with the purified hLRRC23$^{WT}$ or hLRRC23$^{Mut}$ and subjected to immunoblotting. (**F**) Interaction of hLRRC23 to a RS head component, RSPH9. The purified hLRRC23 were incubated with the RSPH-Trap (RS head, RSPH6A and RSPH9; stalk, RSPH3 and RSPH22) and subjected to immunoblotting. 5% amount of the hLRRC23s used for the trap assay were loaded as inputs. Yellow lines in individual α-HA blot images indicate marker information (75 kDa, *left*; 50 kDa, *right*). Experiments were repeated four times. Purified GST was used for negative control (***Figure 3—figure supplement 1B***). Experiments were repeated three times with biological replications. (**G**) A phylogenetic tree constructed by Maximum-likelihood analysis of the protein sequences of the *C. reinhardtii* RSP15 and the orthologs of LRRC23 and LRRC34. LRR37, the first LRRC23 ortholog identified in *Ciona intestinalis* is marked in bold. (**H**) Comparison of the reported RSP15 from *C. reinhardtii* and the predicted structure of LRRC23 and LRRC34 from human. Atomic structure of the *C. reinhardtii* RS2 containing RSP15 are represented by ribbon (RS2) and surface (RSP15) diagram (*left*, PDB Id: 7JU4). Ribbon diagrams of *C. reinhardtii* RSP15 and AlphaFold-predicted human LRRC23 (*middle*) and LRRC34 (*right*) are shown for structural comparison. Secondary structures are color-coded. Different from *C. reinhardtii* RSP15 and LRRC34, LRRC23 does not display repeated α-helix (magenta) between β-sheets (gold).

The online version of this article includes the following source data and figure supplement(s) for figure 3:

**Source data 1.** Uncropped gel and blot images for *Figure 3*.

**Figure supplement 1.** A predicted RSP15 ortholog, *LRRC34*, in metazoan species.

**Figure supplement 1—source data 1.** Uncropped blot images for *Figure 3—figure supplement 1*.

assay (***Figure 3C–E*** and ***Figure 3—figure supplement 1A***). Purified GST-tagged hLRRC23$^{WT}$ and hLRRC23$^{Mut}$ proteins (***Figure 3C and D***) were incubated with a recombinant human RSPH enriched by immunoprecipitation (***Figure 3E***). This trap assay demonstrated that hLRRC23$^{WT}$ interacts only with RSPH9, a RS head protein, among the head and stalk RSPH proteins tested (***Figure 3F*** and ***Figure 3—figure supplement 1B***). Interestingly, the previously reported interaction between LRRC23 and RSPH3 (***Zhang et al., 2021***) is not detected in our assay, which may be due to the different interaction conditions used in vitro. Markedly, hLRRC23$^{Mut}$ does not interact with RSPH9, indicating LRRC23 interaction with RS head via its C-terminus. This result also raises the question whether LRRC23 is a head protein of RS, not a stalk protein, a different picture from previous studies. To test this new hypothesis, we performed BLAST search and found *C. reinhardtii* RSP15 (***Gui et al., 2021***) has the highest sequence homology to LRRC34, not LRRC23, in *Ciona intestinalis*. Our phylogenetic and pairwise distance comparison analyses also revealed that LRRC34 orthologs are evolutionarily closer to RSP15 orthologs than LRRC23 orthologs (***Figure 3G*** and ***Figure 3—figure supplement 1C***). Moreover, AlphaFold-predicted structure of human LRRC34, but not that of LRRC23, presents the same structural features as those of RSP15 (i.e. repeated leucin-rich repeat domains and an α-helix motif in-between) (***Figure 3H***). *LRRC34* and *LRRC23* share their gene expression patterns among tissues, most abundantly expressed in the tissues known for ciliary and flagellar function such as retina, testis, and fallopian tube (***Figure 3—figure supplement 1D and E***). All these results suggest that LRRC34 is a ciliary and flagellar protein and likely the RSP15 orthologue in chordates, and that LRRC23 function is associated with the RS head.

## LRRC23 mutation disorganizes radial spoke 3 in sperm flagella

Next, we examined the impact of LRRC23 loss of function by the C-terminal truncation on the subcellular and ultrastructural organization of sperm. We first compared flagellar compartmentalization between *Lrrc23$^{+/\Delta}$* and *Lrrc23$^{\Delta/\Delta}$* sperm. Confocal imaging of immunostained sperm by antibodies against various proteins of known localization did not show any difference on the subflagellar localization of axonemal or peri-axonemal proteins (***Figure 4A*** and ***Figure 4—figure supplement 1A***). The levels of such flagellar proteins are also not significantly different between *Lrrc23$^{+/\Delta}$* and in *Lrrc23$^{\Delta/\Delta}$* sperm (***Figure 4—figure supplement 1B and C***). Furthermore, transmission electron microscopy (TEM) did not reveal apparent structural abnormalities in *Lrrc23$^{\Delta/\Delta}$* sperm flagella (***Figure 4B*** and ***Figure 4—figure supplement 1D***). These results indicate overall subflagellar and ultrastructural organization in *Lrrc23$^{\Delta/\Delta}$* sperm is preserved despite almost complete loss of sperm motility. Any structural abnormality in *Lrrc23$^{\Delta/\Delta}$* sperm would be subtle and local in the axoneme, likely in the RS head region, which requires a higher resolution microscope technique.

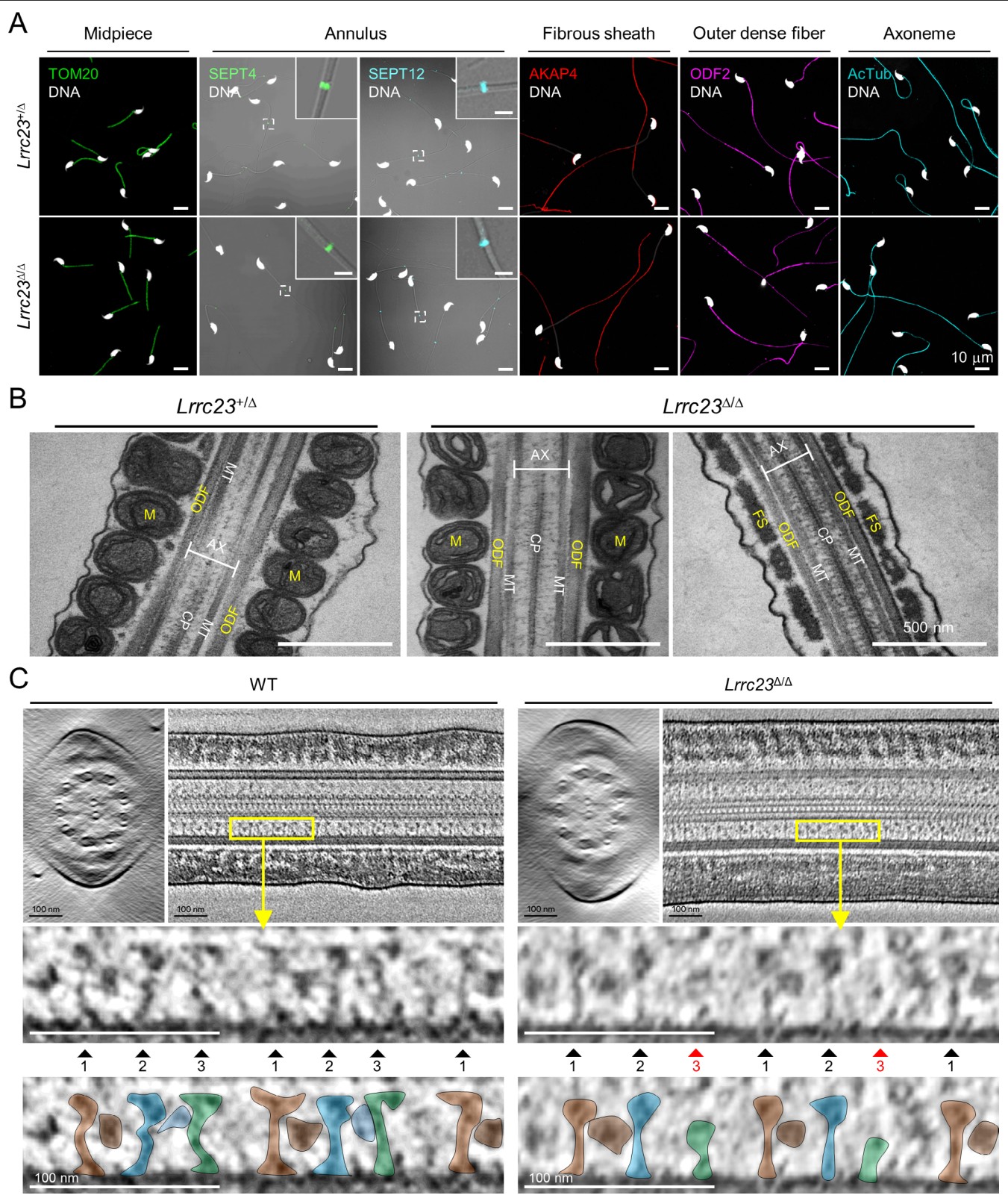

**Figure 4.** LRRC23 mutation disrupts the third radial spoke (RS) in sperm flagellum. (**A**) Immunostaining of flagellar proteins in different compartments. Shown are midpiece (TOM20), annulus (SEPT4 and SEPT12), fibrous sheath (AKAP4), outer dense fiber (ODF2), and axoneme (acetylated tubulin, AcTub) in *Lrrc23*[+/Δ] (*top*) and *Lrrc23*[Δ/Δ] (*bottom*) sperm. Magnified insets are represented for annulus proteins (scale bars in insets = 2 μm). Fluorescence and corresponding DIC images are merged. Sperm heads were counter stained with Hoechst. *Lrrc23*[+/Δ] sperm were used for positive control. Experiments

*Figure 4 continued on next page*

*Figure 4 continued*

were performed with three biological replications. (**B**) Transmission electron microscopy images of *Lrrc23*^+/Δ (*left*) and *Lrrc23*^Δ/Δ (*right*) sperm. Shown are longitudinal section of sperm flagella. M, mitochondria; ODF, outer dense fiber; AX, axoneme; CP, central pair; MT, microtubule; FS, fibrous sheath. *Lrrc23*^+/Δ sperm were used for positive control. (**C**) Cryo-electron tomography (cryo-ET) of WT and *Lrrc23*^Δ/Δ sperm flagella. Shown are representative tomographic slices from WT (*left*) and *Lrrc23*^Δ/Δ sperm (*right*). The 9+2 axonemal structure are shown in both WT and *Lrrc23*^Δ/Δ in cross-sectional view (*left*). Axonemal structures are shown with proximal side of the flagellum on the left in longitudinal view (*right*; see *Video 3*). Magnified insets (*bottom*) reveal that RS1, 2, and 3 are shown in WT sperm (*left*, filled arrowheads) but RS3, especially head part, is not clearly visible (*right*, red arrowheads) in *Lrrc23*^Δ/Δ sperm. RS1, 2, and 3 are distinguished by the interval between each set of RS1, 2, and 3, and the electron dense area corresponding to the barrel (RS1) and bridge (RS2-3) structures. WT sperm were used for positive control.

The online version of this article includes the following source data and figure supplement(s) for figure 4:

**Figure supplement 1.** Flagellar compartmentalization in *Lrrc23*-mutant sperm.

**Figure supplement 1—source data 1.** Uncropped blot images for *Figure 4—figure supplement 1*.

To determine how the absence of LRRC23 affects sperm structure in more detail, we performed cryo-ET 3D reconstruction to visualize substructural changes of the sperm axoneme (*Figure 4C* and *Video 3*). The reconstructed tomogram slices revealed striking details of the RS structural difference between WT sperm and *Lrrc23*^Δ/Δ sperm (*Figure 4C*). In WT sperm, the three RSs are repeated with a typical pattern, in which RS1 and RS2 are recognized by an additional EM density between them (barrel structure, *Leung et al., 2021*) followed by RS3 (*Figure 4C*, *left*). In *Lrrc23*^Δ/Δ sperm, EM densities corresponding to RS3 are significantly weaker than those in WT sperm, whereas the structural features of RS1 and RS2 are overall kept unaltered from that of WT sperm (*Figure 4C*, *right*). These results strongly indicate that the LRRC23 mutation specifically disorganizes RS3 in the sperm axoneme.

## LRRC23 is a head component of radial spoke 3

To visualize the structural defects of RS3 along the *Lrrc23*^Δ/Δ sperm axoneme in more detail (*Figure 4C*), we performed sub-tomogram averaging (STA) of the axonemal doublets in 96 nm repeat (i.e. the distance that spans a single set of RS1, RS2, and RS3) on both WT and *Lrrc23*^Δ/Δ spermatozoa (*Figure 5* and *Figure 5—figure supplement 1*). Remarkably, the resulting 3D maps reveal that the head region of RS3 is entirely missing in *Lrrc23*^Δ/Δ sperm, whereas the heads of RS1 and RS2 are intact (*Figure 5A and B*). In addition, superimposition of the 3D STA maps demonstrates that the junctional structure between RS2 and RS3–present in mouse and human sperm but not in *C. reinhardtii* nor in *T. brucei* flagellar axoneme–is also specifically abolished in *Lrrc23*^Δ/Δ sperm (*Figure 5C*). By contrast, *Lrrc23*^Δ/Δ sperm have intact stalk structures of RS3 like those in WT sperm. Consistent with the protein-protein interaction between LRRC23 and the RS head protein RSPH9 using the RSPH-trap approach (*Figure 3*), these direct structural observations unarguably clarify that LRRC23 is a RS3 head component and the absence of the RS3 head leads to motility defects in *Lrrc23*^Δ/Δ sperm. These results demonstrate that the C-terminal truncation of LRRC23 prevents the assembly of RS3 head during spermatogenesis, thus preventing functional RS3 complex formation. Taken together, our functional and structural studies using the mouse model recapitulating the human mutation elucidate the molecular pathogenicity of LRRC23 underlying impaired sperm motility and male infertility (*Figure 5D*).

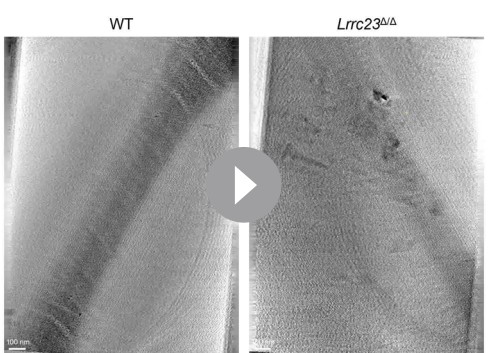

**Video 3.** Tilted series of cryo-electron tomogram slices from WT (*left*) and *Lrrc23*^Δ/Δ (*right*) spermatozoa. Tomogram images were acquired of WT and *Lrrc23*^Δ/Δ sperm were acquired on the grid and image slices were rendered to show the axonemal structure in a tilt series. https://elifesciences.org/articles/90095/figures#video3

## Discussion

### LRRC23 is a distinct head component of radial spoke 3

Accumulating evidence on the structural heterogeneity of radial spokes in cilia and flagella suggests that the molecular organization and function of RS3 is distinct from those of RS1 and RS2. For example, the morphology of RS3 head is distinguished from those of RS1 and 2 in mouse

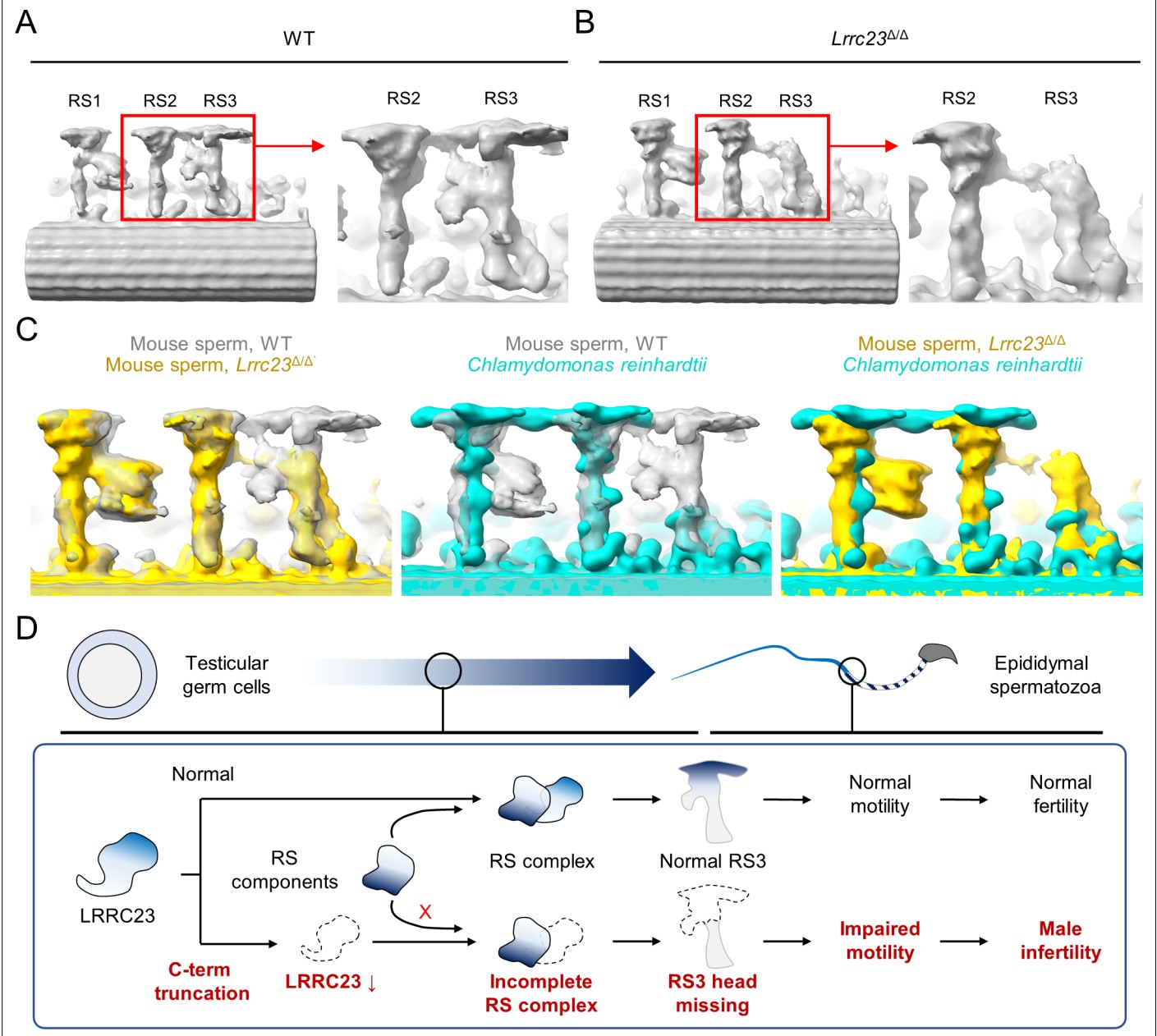

**Figure 5.** Head of the third radial spoke is absent in *Lrrc23*^Δ/Δ sperm flagella. (**A, B**) Sub-tomogram averaging (STA) to analyze structural defects at radial spoke (RS) of WT (**A**) and *Lrrc23*^Δ/Δ sperm (**B**). Shown are STA images resulted from 96 nm doublet repeats from WT and *Lrrc23*^Δ/Δ sperm. RS2 and 3 are magnified and density to represent RS3 head and the bridge between RS2 and RS3 (red circle) is missed in *Lrrc23*^Δ/Δ sperm specifically. (**C**) Overwrapped STA images from 96 nm-doublet repeats from WT (gray) and *Lrrc23*^Δ/Δ (gold) sperm, and *Chlamydomonas reinhardtii* (cyan). (**D**) A proposed model of impaired sperm motility and male infertility by the LRRC23 loss of function.

The online version of this article includes the following figure supplement(s) for figure 5:

**Figure supplement 1.** Workflow of cryo-electron tomography (cryo-ET) and sub-tomogram averaging (STA) processing of 96 nm microtubule doublet repeat from mouse sperm.

motile cilia (*Zheng et al., 2021*). Moreover, in the tracheal cilia of the PCD patients, RSPH1 or RSPH4 loss-of-function mutation specifically abolished RS1 and RS2 heads, but not RS3 head (*Lin et al., 2014*; *Zhao et al., 2021*) even though RS3 shares T-shaped structure just like RS1 and RS2 in most eukaryotic motile cilia and flagella (*Imhof et al., 2019*; *Leung et al., 2021*; *Lin et al., 2012*; *Lin et al., 2014*). Despite these findings, the molecular composition of RS3 head remains largely unknown. The current study demonstrates that LRRC23 is an RS3-specific head component. Previous immuno-EM

studies showed that LRRC23 is a conserved RS protein in *C. intestinalis* and mouse (*Pigino et al., 2011*; *Zhang et al., 2021*). It is of note that LRRC23 was originally predicted to be the orthologue of RSP15 in *C. reinharditti*, a RS2 stalk protein, due to the presence of leucine-rich repeat domains (*Gui et al., 2021*; *Yang et al., 2006*). Yet, we found LRRC23 orthologues are not conserved in *C. reinharditti* in which RS3 is headless and short stump-like, but present in the species where RS3 head structure is preserved, such as *T. thermophila*, sea urchin, and mammals. Instead of LRRC23, our phylogenetic and structural comparison strongly suggests that LRRC34 is likely the RSP15 ortho-logue in chordate animals (*Figure 3*, *Figure 3—figure supplement 1*) and a previously unappreci-ated component of RS2 stalk. Identifying a LRRC34 loss-of-function mutation from ciliopathic children further supports that LRRC34 is a RS component (*Shamseldin et al., 2020*). Our wild-type-mutant comparison approach using cryo-ET analyses ultimately clarify that LRRC23 is required for assembling the RS3 head structure (*Figures 4 and 5*), indicating LRRC23 as an RS3 head component. Interest-ingly, a RS head component, RSPH9, interacts with LRRC23 but the protein level or the localization of RSPH9 is not altered in *Lrrc23*^Δ/Δ sperm (*Figure 4—figure supplement 1*), suggesting that RSPH9 could be a head component of RS1 and RS2 like in *C. reinhardtii* (*Gui et al., 2021*), but not of RS3. Of note is that an independent study reported that LRRC23 is required for flagellar localization of the RS stalk and fibrous sheath components in mature sperm (*Zhang et al., 2021*). This discrepancy between two studies is likely due to the presence of the truncated LRRC23 in testis in *Lrrc23*^Δ/Δ males, which might partly allow flagellar localization of other flagellar components during germ cell development. Although the full picture of molecular composition of RS3 head remains to be revealed, our findings demonstrate LRRC23 is a RS3-specific head component. This conclusion is further supported by the presence of LRRC23 in tracheal cilia (*Zhang et al., 2021*) that lack the RS2-RS3 bridge structure (*Leung et al., 2021*).

## LRRC23 is required for mammalian sperm-specific bridge structure between RS2 and RS3

In motile cilia and flagella, a set of three RSs is repeated along the axoneme (*Leung et al., 2021*; *Lin et al., 2012*; *Lin et al., 2014*; *Viswanadha et al., 2017*). Notably, a recent cryo-ET study revealed additional RS substructures in mammalian sperm flagella, a barrel structure at RS1 and a junctional bridge structure between RS2 and RS3 (*Leung et al., 2021*), which were not observed in the sea urchin sperm or human motile cilia (*Lin et al., 2014*). Furthermore, they are asymmetrically distributed along the mammalian sperm axoneme, corresponding to the peripheral MTDs in a species-specific manner (*Chen et al., 2023*). Our cryo-ET and STA analyses visualized the mammalian sperm-specific RS1 barrel and RS2-RS3 bridge structures in WT sperm flagella, consistent with *Leung et al., 2021*. Strikingly, in *Lrrc23* mutant sperm, most of the EM density corresponding to the RS2-RS3 bridge struc-ture is missing and/or altered together with that of the RS3 head (*Figures 4 and 5*). Considering the absence of RS2-RS3 bridge in tracheal cilia (*Lin et al., 2014*), LRRC23 also contributes to assemble this flagellar RS substructure. Thus, we speculate that the RS3 and RS2-RS3 bridge structures and their sub-axonemal localization confer non-planar and asymmetric flagellar motility unique to mammalian sperm hyperactivation. As the RS2-RS3 bridge structure is absent in tracheal cilia (*Lin et al., 2014*), our study unambiguously demonstrates that LRRC23 is required for assembling this bridge structure specifically in mammalian sperm flagella. If so, LRRC23 may be localized at the junction between RS2-RS3 bridge structure and RS3 head. The detailed molecular components that comprise the RS2-RS3 bridge require further study. Profiling and comparing LRRC23 interactomes from mammalian motile cilia and sperm flagella could unveil the cell-type-specific molecular organization of RS3.

## LRRC23 loss of function causes male infertility in mice and human

Loss-of-function mutations in various RS components that are common to motile cilia and sperm flagella were reported to cause PCD and/or male infertility. Loss-of-function mutations of RSPH1 (*Knowles et al., 2014*; *Kott et al., 2013*), RSPH3 (*Jeanson et al., 2015*), RSPH4A and RSPH9 (*Castleman et al., 2009*) were identified from PCD patients. Some of the male patients carrying RSPH1 and RSPH3 mutations were infertile (*Jeanson et al., 2015*; *Knowles et al., 2014*). RSPH1, RSPH4A, and RSPH9 knockout mouse models recapitulated the PCD phenotypes such as hydrocephalus and impaired mucociliary clearance (*Yin et al., 2019*; *Yoke et al., 2020*; *Zou et al., 2020*). However, there are other RS components in which mutations only cause male infertility in mice and human. For example, WES

of infertile males without PCD symptoms identified mutations in CFAP251 (*Auguste et al., 2018*; *Kherraf et al., 2018*) and CFAP61 (*Liu et al., 2021*; *Ma et al., 2021*). RSPH6 (*Abbasi et al., 2018*), CFAP251 (*Kherraf et al., 2018*), or CFAP61 (*Liu et al., 2021*) deficiency also rendered male mice infertile but without gross abnormalities. These phenotypic differences could be due to a different physiological role of the individual RS component between motile cilia and flagella or a distinct repertoire of the RS components that permits functional redundancy. In our study, LRRC23 mutant mice do not have any apparent gross abnormality but display male infertility (*Figure 2*). Considering the altered interaction of truncated LRRC23 with RSPH9 in vitro (*Figure 3*), the LRRC23 interaction with RSPH9 is presumably essential for the RS organization and/or sperm motility in mammals. However, since we only examined the interaction of the human proteins, there may be species-specific differences that need to be further investigated. Consistent with our study, a previous study found immotile sperm but normal tracheal ciliary beating in the absence of LRRC23 (*Zhang et al., 2021*). Supportive of these observations, the infertile male patients in the current study do not show PCD symptoms. This phenotype of male infertility without PCD symptoms in both mice and human suggests the mammalian sperm-specific role of LRRC23 in RS structure and function. Physiological implication of LRRC23 and RS3 in motile cilia is unclear but it is likely dispensable for normal ciliary movement probably due to compensatory and/or redundant RS3 proteins specific to cilia. Whether LRRC23 absence would lead to similar structural aberration in RS3 head of motile cilia requires further study. Intriguingly, LRRC23 loss-of-function impairs primarily flagellar motility but morphology only marginally, which is in distinct from other RS components. For example, the absence of RSPH6, CFAP251, or CFAP61 causes male infertility without PCD symptoms but displays multiple morphological abnormalities of the flagella characterized by absent, short, bent, coiled, and irregular flagella (*Touré et al., 2021*). By contrast, LRRC23 mutation and absence do not cause either PCD nor MMAF phenotypes in human and mouse, suggesting a distinct physiological significance of LRRC23 in reduced sperm motility.

## Materials and methods

### Subject and family

This study was approved from the review board of Quaid-i-Azam University, Islamabad, Pakistan (IRB00003532, IRB protocol # QAU-171) and the Yale Center for Mendelian Genomics. The family members recruited in this study were explained about the procedure and possible outcomes. The family members provided written consent to attend this study.

### Sample collection and clinical investigation

Semen

Semen samples were collected and clinically analyzed according to WHO guidelines (*World Health Organization, 2010*). Semen samples were collected from the infertile male members after 2–5 days of abstinence from sexual intercourse. The collected semen samples were subjected to the clinical analysis by expert laboratory technologists at Aga Khan Medical Centre (Gilgit, Pakistan) to measure pH, volume, viscosity, and color of the semen and sperm parameters. To assess sperm motility and morphology, the collected semen was liquefied at 37 °C for 30–60 min. A minimum of 1000 spermatozoa were counted to analyze sperm motility by the CASA system using an MTG-GmbH analyzer (version 5.4; MTG-MedeaLAB) according to the unit's standard operating procedures based on the WHO guidelines as previously reported (*Krause et al., 2016*; *Nawaz et al., 2021*; *Slabbert et al., 2015*). Sperm from liquified semen were subjected to PAP-staining and the morphology was microscopically analyzed according to the guidelines from WHO and evaluated by a reproductive medicine specialist. PAP-stained sperm were imaged using CMOS camera (Basler acA1300-200µm, Basler AG) equipped in Nikon E200 microscope.

Blood

Venous blood samples were collected from attending family members. Collected blood samples were used for karyotyping and genomic DNA extraction. Blood cells cultured with phytohemagglutinin for 72 hr were used for karyotyping. The cultured cells were disrupted and Giemsa-stained. Twenty metaphases were examined to examine karyotypes of each member. Genomic DNA (gDNA) samples were extracted using QIAamp DNA Kit (QIAGEN, Germany). Extracted gDNA samples were

subjected to examine microdeletions at the Y-chromosome (AZFa, AZFb, AZFc, and AZFd), whole exome sequencing (WES), and Sanger Sequencing.

## Whole-exome sequencing and data analysis

Whole exome sequencing was carried out as described in our previous study (*Hwang et al., 2021b*). Briefly, 1.0 μg of genomic DNA from proband's blood is fragmented to an average length of 140 bp using focused acoustic energy (Covaris E210). DNA fragments were subjected to exome capturing using the IDF xGen capture probe panel with an additional 'spike-in' of ~2500 region which are total 620 kb of RefGene coding regions. The captured fragments were pair-end sequenced by 101 bp reading using NovaSeq 6000 with S4 flow cell (Illumina). Sequenced reads were aligned to reference human genome (GRCh37/hg19) using the BWA-MEM (*Li, 2013*) and processed to generate variants using GATK v3.4 (*McKenna et al., 2010*; *Van der Auwera et al., 2013*). Variants were annotated with ANNOVAR (*Wang et al., 2010*) and the predicted deleteriousness of non-synonymous variants was determined by MetaSVM (*Dong et al., 2015*). Either loss of function mutations of stop-gains, stop-losses, frameshift indels, and canonical splice-sites or deleterious missense mutations predicted by MetaSVM were considered to potentially damaging. Recessive variants of which MAF values in Genome Aggregation Database (gnomAD), v2.1 (*Lek et al., 2016*; https://gnomad.broadinstitute.org/) are lower than $10^{-3}$ were considered for rare variants. Rare damaging variants were further filtered to exclude false-positives using the follow criteria: (1) PASS for GATK variant quality score recalibration, (2) MAF ≤2.0 x $10^{-5}$ in the gnomAD, (3) DP ≥8 independent reads, (4) GC score ≥20, (5) MQ score ≥40, (6) PLdiff/DP ≥8, and (7) indels in Low Complexity Regions. Co-segregation of candidate variants with the phenotypes were confirmed by genomic DNA PCR and Sanger sequencing. Genomic DNA PCR was performed with using One*Taq* 2 X Master Mix (NEB) and used primer pairs were; Fwd: 5'-GCTGAGCATTTGGAGTGGC-3' and Rev: 5'-CCTGCTAGGTGGCTGTGTAT-3' for *LRRC23*, Fwd: 5'-TGAACCCCTGGCACAACT-3' and Rev: 5'-TTTTTACTCAGCGATACCACATTCACAG-3' for *SCN5A*, Fwd: 5'- TGGCTAAATCCCATCCAGTCC-3' and Rev: 5'- GAGTCTGTCCTTGCCCGTAG-3' for *NOX1*, and Fwd: 5'-GATTGTCATCGCCTTGTTCATC-3' and Rev: 5'-TGTTTTGTGGTGGCACAGTC-3' for *PRRG3*. Amplified PCR products were subjected to Sanger sequencing.

## Kinship analysis

Kinship coefficient was estimated to confirm pedigree information and pairwise relatedness of proband using KING v2.2.4 (*Manichaikul et al., 2010*). Inbreeding coefficient was calculated by homozygosity-by-descent (HBD). HBD segment in the proband was detected by Beagle v3.3.2 (*Browning and Browning, 2011*). Homozygosity in segments over 2 cM was considered for consanguinity.

## Animals

Wildtype C57BL/6 mice were from Charles River Laboratory. Mice were cared in accordance with the guidelines approved by Institutional Animal Care and Use Committee (IACUC) for Yale University (#20079).

## Generation of the *Lrrc23*-mutant mice and genotyping

*Lrrc23*-mutant mice were generated on C57BL/6 background using CRISPR/Cas9 genome editing as described (*Chen et al., 2016*; *Yang et al., 2014*). Two guide RNAs targeting the 5th (5'- CATA TGGTAACATTGACCCAGGG-3') and 7th (5'- CGTCTCTACCAGCTATACAGCGG-3') were in vitro transcribed and purified. The sgRNAs/Cas9 RNPs were complexed and electroporated into zygotes from C57BL/6 J. The embryos were transplanted to oviducts of pseudopregnant CD-1 foster females and founders' toe were biopsied to extract gDNA. Truncation of the target region was confirmed by gDNA PCR with F3 (5'-CACTTTTCCTGCCTCTGTGTCC-3') and R3 (5'-AGCATCTCCCACTTCCTGTGAC-3') primers. The amplicons were Sanger sequenced and founder females with 4126 bp or 4128 bp truncation at the genomic region (*4126del* and *4135del*) were mated with C57BL/6 WT mice to confirm the germline transmission of the alleles. Two mutant mice lines were maintained and genotyped with F3-R3 pairs for the mutant allele and F2 (5'-TTGTGGTGGTGGGGAGATAG-3')-R2 (5'-GTGGTGAT GGACGGGTGT-3') pair for WT allele.

## Mouse sperm preparation

Sperm were collected from cauda epididymis of the adult mice by swim-out in M2 medium (EMD Millipore). To induce capacitation, the epididymal sperm were incubated in human tubular fluid (HTF; EMD Millipore) at $2.0 \times 10^6$ cells/ml concentration for 90 min at 37 °C, 5% $CO_2$ condition.

## Mammalian cell culture and transient protein expression

Human embryonic kidney 293T cells (ATTC) were cultured in DMEM (Gibco) supplemented with 10% FBS (Gibco) and 1 X Pen/Strep (Gibco) at 37 °C, 5% $CO_2$ condition. Cultured cells were transfected with Lipofectamine 2000 (Invitrogen) or polyethylenimine (PEI) to express recombinant protein transiently. Transfected cells were used for co-immunoprecipitation or modified trap-assay.

## RNA extraction, cDNA synthesis, and PCR

Total RNA was extracted from transfected 293T cells and frozen mouse testes using RNeasy Mini kit (QIAGEN). Extracted RNA was used for cDNA synthesis using iScript cDNA Synthesis kit (BIO-RAD) in accordance with manufacturer's instruction. Synthesized testis cDNAs were used for endpoint PCR or quantitative PCR using OneTaq 2 X Master Mix (NEB) or with iTaq Universal SYBR Green Supermix (BIO-RAD), respectively. Primer pairs, F1 (5'-GGCATCTCTCATCCTCGTCT-3') - R1 (5'-AGCCACTC AGGGTGTCAATC-3'), F4 (5'-TTGGCGTCTCAGCACAAAG-3') - R4 (5'-CTCGAAGCTCCAGGGT GT-3'), and F5 (5'-CTGGACCCCGAGAGACTG-3') - R5 (5'-AGTTTTACCCCCGACCTGTG-3') were used for endpoint PCR; E3 (5'-AGCTGGAGGCTAAGGACAGG-3') - E4 (5'-GAGCGGCGATATGTCT GTAA-3') and E7 (5'-GTCAGAGGCTGAAGGAGGAA-3') - E8 (5'- TATCAGTTCTTGGGGCCAGT-3') were used for quantitative PCR. TBP was used to calculate relative transcript levels in mouse testes by $\Delta\Delta$Ct method.

## Antibodies and reagent

Rabbit polyclonal RSPH3B were described in previous studies (*Hwang et al., 2021b*). Rabbit polyclonal anti-LRRC23 (α–118, PA5-63449; α–208, PA5-58095), DNAH1 (PA5-57826), DNAH2 (PA5-64309), and DNAH9 (PA5-45744) were from Invitrogen. Rabbit polyclonal SEPTIN4 (NBP1-90093) antibody was from Novus Biologicals. Mouse monoclonal anti-His-tag (66005–1-Ig) and rabbit polyclonal anti-AKAP3 (13907–1-AP), RSPH9 (23253–1-AP), and RSPH22 (16811–1-AP) were purchased from Proteintech. Rabbit polyclonal TOM20 (sc-11415) was from SantaCruz. Mouse monoclonal anti-acetylated tubulin (AcTub, clone 6-11B-1, 7451), CALM1 (05–173), pTyr (clone 4G10, 05–321), and HA (clone HA-7, H3663) and rabbit polyclonal anti-SEPTIN12 (HPA041128) were from Sigma-Aldrich. Rabbit and mouse monoclonal anti-DYKDDDDK (clone D6W5B, 86861; clone 9A3, 8146) were from Cell Signaling Technology. Rabbit polyclonal ODF2 and AKAP4 were gifted from Dr. Edward M. Eddy. Rabbit anti-RSPH6A sera was gifted from Dr. Masahito Ikawa. HRP-conjugated goat anti-mouse and anti-rabbit IgG were from Jackson ImmunoResearch. Goat anti-rabbit IgG conjugated with Alexa 568, Lectin PNA conjugated with Alexa 647, and Hoechst dye were from Invitrogen.

## Protein extraction, solubilization, and immunoblotting

### Testis

Adult mouse testes were homogenized in 0.32 M using dounce homogenizer followed by centrifugation for 15 min at 4 °C, 1000 x *g* to pellet nucleus and debris. Supernatants was collected and centrifuged at 4 °C, 100,000 rpm for an hour to separate cytosolic and microsome fractions. Volume-equivalented cytosolic and microsome fractions were lysed with 2 X LDS sampling buffer and denatured by boiling at 75 °C with 50 mM dithiothreitol (DTT) for 2 or 10 min, respectively, for SDS-PAGE and immunoblotting. Primary antibodies used for immunoblotting were: rabbit polyclonal anti-LRRC23 (α–118 and α–208, both 1:500) and mouse monoclonal anti-AcTub (1:2000), and calmodulin (1:1000).

### Epididymal sperm

Collected epididymal cells were washed with PBS and lysed using 2 X LDS by vortexing at room temperature (RT) for 10 min to extract whole sperm proteins as described previously (*Hwang et al., 2022*). The lysates were centrifuged for 10 min at 4 °C, 18,000 x *g* and the supernatant were mixed to 50 mM DTT followed by boiling at 75 °C for 10 min for denaturation. The samples were subjected

to SDS-PAGE and immunoblotting. The used primary antibodies were: rabbit polyclonal anti-LRRC23 (α–118 and α–208, 1:500), AKAP3 (1:2000), RSPH9 (1:500), RSPH22 (1:1000), ODF2 (1:2,000), AKAP4 (1:2000), and RSPH3 (1:500), rabbit anti-RSPH6A sera (1:500), and mouse monoclonal AcTub (1:20,000).

## 293T cells

293T cells were lysed using 0.5% Triton X-100 in 10 mM HEPES, 140 mM NaCl, pH7.4 buffer (1 X HEPES buffer) for 2 hours at 4 °C with gentle rocking. The lysates were centrifuged at 4 °C, 18,000 x *g* for 30 min. Supernatant with solubilized proteins were used for protein interaction tests or mixed to 1 X LDS and 50 mM DTT followed by boiling at 75 °C for 2 min. Denatured samples were used for immunoblotting. Mouse monoclonal anti-HA (1:4000) and FLAG (1:1000), and rabbit monoclonal anti-FLAG (1:1000) were used for primary antibody.

HRP-conjugated goat anti-rabbit and goat anti-mouse IgG were used for secondary antibodies (0.1 μg/ml). SuperSignal Western Blot Enhancer (Thermo Fisher Scientific) was used for testis and sperm immunoblotting with anti-LRRC23 antibody (α–118).

## Molecular cloning

Human *LRRC23* ORF clone (HG24717-UT, SinoBiological) and the partial region of the human *LRRC23* 5th intron amplified by PCR was subcloned into phCMV3 vector to generate mammalian expression constructs for human *LRRC23* (*phCMV3-FLAG-hLRRC23$^{ORF}$-HA*, *phCMV3-FLAG-hLRRC23$^{WT}$-HA,* and *phCMV3-FLAG-hLRRC23$^{Mut}$-HA*). cDNA clones of human *RSPH3* (616166, Horizon Discovery), *RSPH6A* (5270908, Horizon Discovery), *RSPH9* (5296237, Horizon Discovery), and *RSPH22* (OHu31347, GenScript) was subcloned into *phCMV3* to express human RSPH proteins tagged with FLAG at C-termini. A stop codon was placed at the upstream of sequences encoding HA in phCMV3 vector for tagging FLAG at C-termini. ORFs encoding full-length and the predicted mutant LRRC23 were amplified from *phCMV3-FLAG-hLRRC23$^{ORF}$-HA* constructs and subcloned into *pGEX-6P2* vector to generate *pGEX-6P2-hLRRC23$^{WT}$* and *pGEX-6P2-hLRRC23$^{Mut}$* constructs tagging with HA at C-termini. ORFs for each construct were amplified using Q5 Hot Start High-Fidelity 2 X Master Mix (NEB) and subcloned into linear vectors using NEBuilder HiFi DNA Assembly Kit (NEB).

## Recombinant protein purification

Bacterial expression constructs were transformed to BL21-CodonPlus(DE3)-RIL competent cells (Agilent Technologies). Fresh colonies were cultured into LB with antibiotics overnight at 37 °C and cultured further after 50 times dilution at 37 °C until OD$_{600}$ values reach to 0.5–0.8. The cultivates were treated with 1 mM IPTG to express recombinant proteins and cultured further for 16–18 hr at 16 °C. IPTG-treated bacteria were washed with PBS and eluted with 1 X HEPES buffer containing 1% Triton X-100 and EDTA-free protease inhibitor cocktail (Roche) The elutes were sonicated and centrifuged at 18,000 x *g*, 4 °C for 1 hr. Supernatant were collected and incubated with glutathione agarose (Pierce) for overnight at 4 °C to purify recombinant GST and GST-tagged human LRRC23$^{WT}$ and LRRC23$^{Mut}$. The incubated resins were washed with PBS and eluted with 1 X HEPES buffer supplemented with 10 mM reduced glutathione. The elutes were dialyzed against 1 X HEPES buffer with 50% glycerol for overnight at 4 °C. Purified proteins were subjected to Coomassie gel staining using Imperial Protein Stain (Thermo Scientific) and immunoblotting.

## Modified trap assay

293T cells to express FLAG-tagged human RSPH proteins transiently were lysed with 1% Triton X-100 in 1 X HEPES buffer with EDTA-free protease inhibitor cocktail (Roche) by rocking at 4 °C for 2 hours. The lysates were centrifuged at 18,000 x *g* for 30 min at 4°C and the supernatant was incubated with Surebeads Protein A Magnetic Bead (Bio-Rad) conjugated with rabbit monoclonal DYKDDDDK antibody at RT for 2 hr. The magnetic beads were washed with 1% Triton X-100 in 1 X HEPES buffer two times and 0.2% Triton X-100 in 1 X HEPES buffer. Purified GST, GST-tagged human LRRC23$^{WT}$ and LRRC23$^{Mut}$ proteins were incubated with the washed magnetic beads in 0.2% Triton X-100 in 1 X HEPES buffer at 4 °C for overnight. Incubated magnetic beads were washed with 0.2% Triton X-100 in 1 X HEPES buffer for three times and eluted with 2 X LDS buffer containing 50 mM DTT followed by denatured 75 °C for 10 min.

## Sperm fluorescence staining

Epididymal sperm cells were washed with PBS and attached on the glass coverslips by centrifugation at 700 x $g$ for 5 min. The coverslips were fixed with either 4% PFA in PBS at RT for 10 min (SEPTIN4, SEPTIN12, RSPH3, DNAH2, DNAH9, TOM20, and LRRC23) or acetone at – 20 °C for 5 min (RSPH9, DNAH1, AKAP4, ODF2, and AcTub). PFA fixed coverslips were washed with PBS three times and permeablized with 0.1% (SEPTIN4 and SEPTIN12), 0.5% (RSPH3, LRRC23, and DNAH9), or 1% (DNAH2) Triton X-100 in PBS at RT for 10 min. Acetone-fixed coverslips were rehydrated by washing with 0.1% Triton X-100 in PBS and PBS. Permeablized coverslips were blocked with 10% normal goat serum in PBS and incubated with primary antibodies in blocking buffer at 4 °C overnight. Used primary antibodies were: Rabbit polyclonal anti-LRRC23 (α–118, 1:100), SEPTIN4 (1 µg/ml), SEPTIN12 (1:100), RSPH3 (10 µg/ml), RSPH9 (1:100), DNAH1 (3 µg/ml), DNAH2 (0.5 µg/ml), DNAH9 (5 µg/ml), TOM20 (1:50), AKAP4 (1:100), and ODF2 (1:50), and mouse monoclonal anti-AcTub (1:200). Coverslips were washed with 0.1% Triton X-100 in PBS one time and PBS two times and incubated with either goat anti-rabbit or mouse IgG conjugated with Alexa 568 in blocking buffer at RT for an hour. The coverslips were washed with PBS three times and mounted on the glass slide using Vectasheild (Vector Laboratory). To observe sperm acrosome, PFA-fixed coverslips were incubated with PNA conjugated with Alexa 647 at RT for an hour. Fluorescence stained coverslips were imaged using Zeiss LSM710 Elyra P1 with Plan-Apochrombat 63 X/1.40 objective (Carl Zeiss). Hoechst were used for counter staining.

## Mating test

Adult heterozygous and homozygous *Lrrc23* mutant male mice were housed with adult female mice with normal fertility and monitored over two months. The pregnancy and litter size were recorded.

## Sperm motility analysis

### Computer-assisted sperm analysis

Computer-assisted sperm analysis (CASA) was performed as previous study (*Hwang et al., 2022*). Briefly, sperm cells at $3.0 \times 10^6$ cells/ml concentration were loaded in slide chamber (CellVision) and their motility parameters were measured on 37 °C warm-stage. Motility of over 200 sperm was recorded using CMOS video camera (Basler acA1300-200µm, Basler AG) at 50 frame per seconds (fps) through 10 x phase contrast objective (CFI Plan Achro 10 X/0.25 Ph1 BM, Nikon) equipped in Nikon E200 microscope. The recorded sperm motility was analyzed using Sperm Class Analyzer software (Microptic).

### Sperm swimming path analysis

Sperm free swimming analysis was conducted as previous study (*Hwang et al., 2019*; *Hwang et al., 2021a*). Briefly, sperm cells were transferred to Delta-T culture dish controller (Bioptech) filled with 37 °C HEPES-buffered HTF medium with 0.3% methylcellulose (*Chung et al., 2017*). Sperm swimming was imaged using pco.edge sCMOS camera in Axio observer Z1 microscope (Carl Zeiss) for 2 s with 100 fps. FIJI software (*Schindelin et al., 2012*) was used to generate overlaid images to show sperm swimming paths.

### Flagellar waveform analysis

To analyze sperm flagellar waveform, $2.0 \times 10^5$ cells of capacitated and non-capacitated sperm cells were placed into 37 °C HEPES-buffered HTF medium (*Chung et al., 2017*) in fibronectin-coated Delta-T culture dish controller (Bioptech). Flagellar movement of the head-tethered sperm was recorded using pco.edge sCMOS camera equipped in Axio observer Z1 microscope (Carl Zeiss) for 2 s with 200 fps. Recorded image stacks were applied to generate overlaid images to show flagellar waveform for two beating cycles with FIJI software (*Schindelin et al., 2012*).

## Sequence alignment and phylogenetic analysis

Amino acid sequences of human and mouse normal and mutant LRRC23 were aligned using Clustal Omega (https://www.ebi.ac.uk/Tools/msa/clustalo/). Amino acids sequences of LRRC23 and LRRC34 orthologs and *Chlamydomonas reinhardtii* RSP15 were aligned with ClustalW for phylogenetic analysis using MEGA7 (*Kumar et al., 2016*). Pairwise distances between the sequences were calculated

with default option. An unrooted phylogenetic tree was generated by maximum-likelihood analysis with 500 bootstrap replications.

### Protein structure analysis

Radial spoke 2 and RSP15 structures of the *C. reinhardtii* were from RCSC Protein Data Bank, PDB (https://www.rcsb.org/; 7JU4). Predicted structures of human LRRC23 (Q53EV4) and LRRC34 (Q8IZ02) were from AlphaFold Protein Structure Database (*Jumper et al., 2021*; https://alphafold.ebi.ac.uk/). Protein structures were rendered and visualized by Mol* 3D Viewer at RCSC PDB.

### Tissue and testicular expression analysis

Transcript expression data of *LRRC23* and *LRRC34* in human tissues was obtained from GTEx and is based on The Human Protein Atlas version 21.0 and Ensembl version 103.38 (https://www.protein-atlas.org/about/download). Medians of the normalized transcript per million (nTPM) are used to calculate relative tissue expression levels. Relative tissue expression levels of *LRRC23* and *LRRC34* are represented as heatmap using GraphPad Prism 8.0 (GraphPad Software Inc, San Diego, CA). UMAP images for the single-cell expression of *LRRC23* and *LRRC34* in human testis are obtained from UCSC Cell browser (*Soraggi et al., 2021*; https://cells.ucsc.edu/?ds=testis).

### Transmission electron microscopy

Transmission electron microscopy was performed as previous study (*Hwang et al., 2021b*). Washed epididymal sperm were pelleted by centrifugation and fixed with 2.5% glutaraldehyde and 2% para-formaldehyde in 0.1 M cacodylate buffer pH7.0 for 30 min at RT and for 1 hr at 4 °C. The fixed pellets were briefly washed with 0.1 M cacodylate buffer pH7.0 and placed in 2% agar. The chilled blocks were trimmed and rinsed with 0.1 M cacodylate buffer followed by placed in 0.1% tannic acid in 0.1 M cacodylate buffer for 1 hr. The samples were washed and post-fixed in 1% osmium tetroxide and 0.15% potassium ferrocyanide in 0.1 M cacodylate buffer for 1 hr. Post-fixed samples were rinsed with the cacodylate buffer and distilled water and subjected to en bloc staining in 2% aqueous uranyl acetate for 1 hr. The samples were dehydrated with ethanol series and infiltrated with epoxy resin Embed 812 (Electron Microscopy Scienes). The resins were placed in silicone molds and backed for 24 hr at 60 °C. The blocks were sectioned in 60 nm depth using Leica UltraCut UC7 and the sections were collected on the formvar/carbon-coated grids. Sections on the grids were stained using 2% uranyl acetate and lead citrate. The stained grids were imaged using MORADA CCD camera (Olympus) equipped in FEI Tecnai Biotwin Transmission Electron Microscope (FEI, Hillsboro, OR) at 80 KV.

### Cryo-electron tomography

#### Grid preparation

Grids for sample imaging were prepared as a previously report (*Leung et al., 2021*). The Quantifoil R2/2 gold grids were glow discharged for 30 s at 15mA and $1.5 \times 10^5$ cells of WT or *Lrrc23*$^{\Delta/\Delta}$ sperm were loaded on the grids followed by incubation at RT for 1 min. The grids were blotted manually from the back site for ~4 s using a manual plunger freezer and immediately plunged into liquid ethane and stored in liquid nitrogen.

#### Cryo-ET data collection

Prepared grids with proper ice particles were screened using a Glacios microscope (200 KV, Ther-moFisher Scientific) at the Yale Science Hill Electron Microscopy Facility. The screened grids were transferred to the Titan Krios microscope (300 KV, Thermo Fisher Scientific) equipped with a Bioquantum Energy Filter and a K3 direct electron detector (Gatan) for data collection at the Yale West Campus Electron Microscopy Facility. Data was collected automatically using SerialEM (*Mastronarde, 2005*). All images were recorded under super-resolution mode with the physical pixel size 3.4 Å (1.7 Å for super resolution movies). 20 and 60 tilt series of WT and *Lrrc23*$^{\Delta/\Delta}$ mouse sperm were collected with the Volta phase plate at a target defocus around –1 μm. Grouped dose-symmetric scheme spanning from –51° to 51° at 3° increment was applied for tilt series acquisition, with a total dose at 100e-/Å2.

#### Tomogram reconstruction

Process of tomogram reconstruction was streamlined using in-house scripts. The movie frames were aligned first using MotionCorr2 and the micrographs were binned with a factor two (*Zheng et al.,*

*2016*). Tilt series stack was generated using in-house script. The tilt-series was aligned by AreTomo 1.0.6 (*Zheng et al., 2022*) to generate XF files which are comparable for IMOD (*Kremer et al., 1996*). The initial CTF parameters were estimated using Ctffind4 (*Rohou and Grigorieff, 2015*). The tomograms were then reconstructed with a binning factor six with 20.4 Å pixel size by IMOD with SIRT algorithm for visualization, particle picking and initial sub-tomogram averaging.

## Sub-tomogram averaging of 96-nm axonemal doublet repeat

### Initial alignment using PEET

MTDs were manually traced in IMOD (*Kremer et al., 1996*). After manual tracing, a polynomial function with degree five was fitted to each microtubule doublet with 24 nm of a sampling distance (Fig EV6C). The polarity of each tomogram is manually determined to reduce the error during subsequent alignment. The 24 nm repeat particles were first aligned using PEET Version 1.15.0 under 'Particle model points' and 'align particle Y axes' options (*Heumann et al., 2011*; *Nicastro et al., 2006*). A published map of 96 nm MTD repeat in WT mouse sperm (EMD-12133; *Leung et al., 2021*) was low-pass filtered to 60 Å and used as the initial reference. A mask covering two adjacent MTDs was generated with 160x160 x 160 nm$^3$ dimension of sub-tomogram particle. Only 'Phi' angle and translational shifts were searched during the alignment. After the initial alignment, positions of the particles were placed at center based on the translational shifts. The Euler angles were extracted and transformed to RELION Euler angle convention (ZYZ) using the MOTL2Relion command in PEET. In-house script was used to gather the coordinates and corresponding Euler angle information for subsequent RELION4 sub-tomogram averaging analysis.

### Refinement and classification

After the metadata preparation, the sub-tomogram particles were made in RELION4 (*Zivanov et al., 2022*) with the binning factor of six. Local 3D refinement with a mask covering one 96 nm MTD was performed. After refinement, the features pf 24 nm repeat of outer dynein arm was observed (Fig EV6D). A smaller mask only covering the inner dynein arm and radial spoke was created for the focused classification without alignment (Fig EV6E). After classification, equivalent 96 nm MTD repeat classes were separated (Fig EV6E). One of the classes was selected for the future process. New sub-tomogram particles with binning factor three and corresponding 10.2 Å of pixel size were generated. Another local refinement was performed with a 96 nm MTD mask. Resolution was estimated using the Fourier shell correlation (FSC) at a cut-off of 0.143 in RELION4 (*Zivanov et al., 2022*). Details of acquisition parameters and particle numbers are summarized in *Supplementary file 3*.

### Visualization

Images for cryo-ET and sub-tomogram averaging were rendered using IMOD (*Kremer et al., 1996*) and UCSF Chimerax (*Goddard et al., 2018*).

## Statistical analysis

Statistical analyses were performed using Student's t-test or Mann-Whitney U test. Significant differences were considered at $*p \leq 0.05$, $**p < 0.01$, and $***p < 0.001$.

## Materials availability

Requests for resources and materials should be directed to the lead contact, Jean-Ju Chung (jean-ju.chung@yale.edu).

## Adherence to community standards

ARRIVE and ICJME guidelines were followed for this work.

## Acknowledgements

The authors highly appreciate participation of the family members in the study presented here. We also thank Muhammad Umair, Khadim Shah, Imran Ullah, Hammal Khan for their help to visit the family for interviews and participation in clinical assessment, Habibur Rehmen and Rina Raza from Aga Khan Medical Centre for clinical semen analyses, Jong-Nam Oh for preparing samples for WES, Case Porter

and Miriam Hill for their help in PCR and Sanger sequencing, the Yale Center for Cellular and Molecular Imaging for assistance in transmission electron microscopy, Dr. Masahito Ikawa from Osaka University for sharing anti-RSPH6A sera. This study was supported by start-up funds from Yale University School of Medicine and National Institute of Child Health and Human Development (R01HD096745) to J-JC; start-up funds from Yale University, National Institute of General Medical Sciences (R35GM142959) to KZ; Pakistan Academy of Sciences (PAS-171) to WA; National Human Genome Research Institute (UM1HG006504) to the Yale Center for Mendelian Genomics. JH was in part supported by Postdoctoral Fellowship from MCI. SN was supported by Pakistan Higher Education Commission International Research Support Initiative Program. JC was in part supported by a Korea University Medical Center Grant. The Genome Sequencing Program Coordinating Center (U24 HG008956) contributed to cross-program scientific initiatives and provided logistical and general study coordination.

## Additional information

### Competing interests

Jungmin Choi, Francesc Lopez-Giraldez, Jean-Ju Chung: Reviewing editor, *eLife*. The other authors declare that no competing interests exist.

### Funding

| Funder | Grant reference number | Author |
|---|---|---|
| Yale University | Start-up funds | Kai Zhang |
| National Institutes of Health | R01HD096745 | Jean-Ju Chung |
| National Institutes of Health | R35GM142959 | Kai Zhang |
| Pakistan Academy of Sciences | PAS-171 | Wasim Ahmad |

The funders had no role in study design, data collection and interpretation, or the decision to submit the work for publication.

### Author contributions

Jae Yeon Hwang, Conceptualization, Data curation, Formal analysis, Validation, Investigation, Visualization, Methodology, Writing – original draft, Writing – review and editing; Pengxin Chai, Data curation, Formal analysis, Methodology, Writing – original draft; Shoaib Nawaz, Formal analysis, Investigation, Methodology; Jungmin Choi, Francesc Lopez-Giraldez, Data curation, Formal analysis, Methodology; Shabir Hussain, Investigation; Kaya Bilguvar, Shrikant Mane, Formal analysis; Richard P Lifton, Resources; Wasim Ahmad, Conceptualization, Resources; Kai Zhang, Resources, Formal analysis, Funding acquisition, Writing – review and editing; Jean-Ju Chung, Conceptualization, Resources, Data curation, Formal analysis, Supervision, Funding acquisition, Writing – original draft, Project administration, Writing – review and editing

### Author ORCIDs

Jae Yeon Hwang  https://orcid.org/0000-0002-6493-4182
Jungmin Choi  http://orcid.org/0000-0002-8614-0973
Jean-Ju Chung  https://orcid.org/0000-0001-8018-1355

### Ethics

Human subjects: This study was approved from the review board of Quaid-i-Azam University, Islamabad, Pakistan (IRB00003532, IRB protocol # QAU-171) and the Yale Center for Mendelian Genomics. The family members recruited in this study were explained about the procedure and possible outcomes. The family members provided written consent to attend this study.
Wildtype C57BL/6 mice were from Charles River Laboratory. Mice were cared in accordance with the guidelines approved by Institutional Animal Care and Use Committee (IACUC) for Yale University (#20079).

Reviewer #1 (Public Review): https://doi.org/10.7554/eLife.90095.3.sa1
Reviewer #2 (Public Review): https://doi.org/10.7554/eLife.90095.3.sa2
Author Response https://doi.org/10.7554/eLife.90095.3.sa3

## Additional files

### Supplementary files

- Supplementary file 1. Clinical diagnosis of the infertile patients.
- Supplementary file 2. Variant Detail in the infertility Family.
- Supplementary file 3. Summarized imaging acquisition parameters and 3D refinement statistics.
- MDAR checklist

### Data availability

The structures for radial spokes from WT (EMD-29013) and LRRC23$^{\Delta/\Delta}$ (EMD-28606) sperm resolved by cryo-ET and STA are deposited to Electron Microscopy Data Bank (https://ebi.ac.uk/pdbe/emdb). All other data generated or analyses during this study are included in the manuscript and supporting files.

The following datasets were generated:

| Author(s) | Year | Dataset title | Dataset URL | Database and Identifier |
|---|---|---|---|---|
| Hwang JY, Chai P, Nawaz S, Choi J, Lopez-Giraldez F, Hussain S, Bilguvar K, Mane S, Lifton RP, Ahmed W, Zhang K, Chung J-J | 2023 | 96nm doublet microtubule repeat from LRRC23-KO mouse sperm | https://www.ebi.ac.uk/emdb/EMD-28606 | Electron Microscopy Data Bank, EMD-28606 |
| Hwang JY, Chai P, Nawaz S, Choi J, Lopez-Giraldez F, Hussain S, Bilguvar K, Mane S, Lifton RP, Ahmad W, Zhang K, Chung J-J | 2023 | 96nm doublet microtubule repeat from wild type mouse sperm | https://www.ebi.ac.uk/emdb/EMD-29013 | Electron Microscopy Data Bank, EMD-29013 |

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
