## [Editor Report · eLife assessment]

This study provides **valuable** findings on a causative relationship between LRRC23 mutations and male infertility due to asthenozoospermia. The evidence supporting the conclusions is **solid**. This work will be of interest to biomedical researchers who work on sperm biology and non-hormonal male contraceptive development.

---

## [Referee Report · Reviewer #1 (Public Review)]

This study identified the truncating LRRC23 is associated with the asthenozoospermia in human and demonstrated that the truncated Lrrc23 specifically disorganizes RS3 and the junctional structure between RS2 and RS3 in the sperm axoneme, which might cause sperm motility defects and male infertility. Although LRRC23 has been reported as a component of the radial spoke and is necessary for sperm motility in mice, this study provided a precise pathogenic mechanism of truncating LRRC23 in asthenozoospermia. This work is of interest to researchers working on reproduction biology. The manuscript has been revised to address prior reviewers' comments.

---

## [Referee Report · Reviewer #2 (Public Review)]

Summary:

The present study explores the molecular function of LRRC23 in male fertility, specifically in the context of the regulation of spermiogenesis. The author initiates the investigation by identifying LRRC23 mutations as a potential cause of male sterility based on observations made in closely related individuals affected by asthenozoospermia (ASZ). To further investigate the function of LRRC23 in spermatogenesis, mutant mice expressing truncated LRRC23 proteins are created, aligning with the identified mutation site. Consequently, the findings confirm the deleterious effects of LRRC23 mutations on sperm motility in these mice while concurrently observing no significant abnormalities in the overall flagella structure. Furthermore, the study reveals LRRC23's interaction with the RS head protein RSPH9 and its active involvement in the assembly of the axonemal RS. Notably, LRRC23 mutations result in the loss of the RS3 head structure and disruption of the RS2-RS3 junction structure. Therefore, the author claimed that LRRC23 is an indispensable component of the RS3 head structure and suggests that mutations in LRRC23 underlie sterility in mice.

Strengths:

The key contribution of this article lies in confirming LRRC23's involvement in assembling the RS3 head structure in sperm flagella. This finding represents a significant advancement in understanding the complex architecture of the RS3 structural complex, building upon previous studies. Moreover, the article's topic is interesting and originates from clinical research, which holds significant implications for potential clinical applications.

---

## [Author Response]

The following is the authors’ response to the original reviews.

The authors deeply appreciate the reviewer’s constructive criticism.

**Answers to the public review from Reviewer 1**
1. The pathogenesis of truncating LRRC23 in asthenozoospermia needs to be further considered. The molecular mechanism of LRRC23 demonstrated in mice should be tested in patients with the LRRC23 variant. As it may be difficult to determine the structures of RS3 in the infertile male sperm, the LRRC23 localization should be observed in the sperm from patients with the LRRC23 variant.

We understand the reviewer’s point. Unfortunately, the patients declined to continue in the project after the initial clinical evaluation and blood draw, so we were unable to follow up.

1. The absence of the RS3 head in LRRC23Δ/Δ mouse sperm is not sufficient to support the specific localization of LRRC23 in RS3 head. Although LRRC23 might bind to RS head protein RSPH9, the authors state that "RSPH9 is a head component of RS1 and RS2 like in *C. reinhardtii* (Gui et al, 2021), but not of RS3" as the protein level and the localization of RSPH9 is not altered in LRRC23Δ/Δ sperm. Thus, the specific localization of LRRC23 in RS3 head should be further confirmed.

Thank you for your comment. We agree with the reviewer that the specific localization of LRRC23 within the RS3 head needs to be further confirmed, but this requires an atomic resolution structure of the RS3 head, which is beyond the scope of the current study. We will pursue this direction in our future study.

1. The interaction between LRRC23 and RSPH9 needs to be defined. AlphaFold models could help determine the likelihood of a direct interaction. In addition, the structure of the 96-nm modular repeats of axonemes from the flagella of human respiratory cilia has been determined (PMID: 37258679), and the localization of LRRC23 in RS could be further predicted.

We appreciate the comment. We are pursuing an atomic resolution structure of the RS3 head, and thus leave the prediction and detailed localization to future studies.

1. The ortholog of the RSP15 may also be predicted or confirmed by using the reported structure in human respiratory cilia (PMID: 37258679). Whether the LRCC34 in RS2 is LRRC34?

Based on the amino acid sequence and AlphaFold predicted structure comparison, we proposed LRRC34 as the RSP15 orthologue. We agree that further clarification of whether the reported RSP15 structure in human respiratory cilia is LRRC34 is valuable, but we would like to focus the current study on re-annotating LRRC23 function to RS3 and male infertility.

**Answers to the public review from Reviewer 2**
1. While the author generated mutant mice expressing truncated LRRC23 proteins, the expression of these truncated proteins was not detected in sperm. This implies that, in terms of sperm structure, the mutant LRRC23 protein behaves similarly to the complete knockout of the LRRC23 protein, which has been previously reported and characterized (Zhang et al., 2021).

We partially agree with the reviewer’s comments. Indeed, the spermatozoa from truncated mutant LRRC23 mice may be similar to those from the complete knockout. However, the truncated LRRC23 in the testis could in part contribute to the assembly and structural organization of the RS3 head and/or bridge during spermatogenesis, and thus it is possible that complete absence of the LRRC23 could result in more severe structural defects in the RS3 and bridge structure. Therefore, to simply infer the same defects requires a direct comparison.

1. This reviewer questions the proposal that LRRC23 is an integral component of RS3, as the results indicate not only the loss of the RS3 head structure but also an incomplete RS2-RS3 junction structure. In addition, the interaction of LRRC23 with RSPH9 alone does not fully explain its involvement solely in RS3 assembly. Additional evidence is required to examine the influence of LRRC23 on the RS2-RS3 junction.

Thank you for the reviewer’s point. In a previous study, LRRC23 was detected in tracheal cilia that lack the bridge structure. Thus, we concluded that LRRC23 is a component in the RS3 head, but not necessarily in the RS2-RS3 bridge structure, although the bridge structure is also affected. Broad structural defects due to single protein loss of function are often observed in sperm flagella. For example, deficiency of RSPH6A, an RS head component, affects not only the RS structure but the entire flagellar structure in a non-uniform manner, resulting in multiple morphological flagellar abnormalities. We anticipate that our future study to determine the molecular architecture in the RS3 head and bridge structure will provide further insights into this question.

1. The article does not explore how these mutations affect the flagella structure in human sperm, which needs further study. Expanding the study to include human sperm structure would undoubtedly enhance the quality of the article.

We agree with the importance of further pursuing the effect of these mutations in human samples, but faced practical difficulties. As responded to reviewer 1, the patients not only dropped out of the project, but also are distantly located in remote region of Pakistan, making the application of cryo-ET not feasible.

**Answers to the recommendations of Reviewer 1**
1. The statistics analysis should be performed in Figures 2E and 2F.

We appreciate the reviewer’s recommendation. For 2E, since the standard deviations for two groups are equal to 0, it is not possible to perform appropriate statical analyses. For 2F, since the knockout males do not sire, it is not possible to know the number of litters in this case. Therefore, litter size information is not available for knockout males, and statistical analyses are not applicable.

1. In Figure 3A, the human sperm RS structures (PMID: 36593309) should be provided.

Thanks for the suggestion. We have included human sperm RS structures as suggested.

1. The molecular weight markers should also be added in Figure 3F (left), EV4B, and EV5B (AKAP3, RSPH9, AcTub).

In the original Figure 3F, the markers were shown as the white lines in the blot images due to the space limitations. Since the previous markers are not clearly visible, we have changed the color to yellow. The marker information in EV4B and 5B has also been updated.

**Answers to the recommendations of Reviewer 2**
1. Line 119, Table S1 is incorrectly shown.

We have corrected the Table nomenclature to Table EV1.

1. Line 132, the author suggests that LRRC23 mutations do not affect female reproduction based on the fertility of the mother. However, this conclusion may lack rigor since it overlooks the sterility of IV-4. To address this, the author needs to examine the fertility of female mice more comprehensively. Additionally, considering the higher expression level of LRRC23 in the oviduct, the author should investigate any potential changes in the oviduct cilia.

Thank you for the reviewer’s comment. As described in line 134, the mother of IV-4, who also carries the homozygous mutant allele like IV-4, was fertile. In addition, Lrrc23Δ/Δ female mice are fertile (now added in lines 173-174). In fact, we maintain the mouse line by crossing Lrrc23Δ/Δ females with heterozygous males. Thus, our initial conclusion that the LRRC23 mutation does not cause female fertility is still valid. However, LRRC23 has a function in the regulation of oviductal cilia requires further study, so we have softened down the corresponding sentence.

1. In the article, the author mentions that there are some morphological differences observed in the sperm, which are not clearly depicted in Fig.1B. It is essential to specify the specific changes in sperm morphology that the author identified.

Thank you for your comment. The morphological variations (e.g., the sperm in the lower left corner of Fig.1B has more a rounded sperm head) meant overall normal morphology with the normal range of occurrence in abnormal sperm morphology in normal fertile men, not necessarily caused by the LRRC23 mutation. To avoid confusion, we have rephrased the sentence (see lines 122-124).

1. In Fig.3F, the previous study confirmed an interaction between LRRC23 and RSPH3 (Zhang et al., 2021), but the current manuscript does not demonstrate such an interaction; the author should explain the text.

We appreciate your point. This could be due to the different interaction condition in vitro, and we described the possibility in main text (See Lines 200-201).

1. In the case of the interaction between LRRC23 and RSPH9, the author utilizes human protein to detect but conducts phenotype verification in mice. Thus, discussing the relevance and potential limitations of extrapolating these findings from human protein interactions to the phenotypic effects

Thank you for the reviewer’s suggestion. We added discussion for that part (lines 336-341).

1. The authors needed to detect changes in LRRC23 protein and mRNA levels at different stages of spermatogenesis.

We agree that expression profiling of LRRC23 protein levels in developing male germ cells will be helpful to further understand LRRC23 function in spermatogenesis, but we do not perceive that it is not critical in this study as LRRC23 mRNA expression profiling from scRNA database (Fig. EV4) hints at the protein profiles.

1. In Figure 4C of the article, the author should provide a clear and detailed explanation in the text of how they distinguish RS1, RS2, and RS3.

We added the information in figure legends (lines 1034-1037).

1. Zoom in on the RS structure in Fig.EV5D for precise observation.

In TEM images with limited resolution, we could not tell which RS (RS1, 2, or 3) we have in the cross-section, and simple zoom-in does not provide a better and/or more accurate observation (the main reason, we moved forward with cryo-ET).

1. By utilizing computational modeling and bioinformatics tools, the authors gain insights into the potential interactions, binding sites, and structural features of LRRC23 within the RS3 complex. This approach provides a deeper understanding of LRRC23's function and role in the assembly and stability of the RS3 complex. To enhance the clarity and visualization of the findings, the authors should generate a schematic diagram that illustrates the proposed interactions and structural organization of LRRC23 within the RS3 complex.

We appreciate the reviewer’s suggestion to speculate the molecular position and interaction of LRRC23 within the RS3 complex. For the level of computational modeling and bioinformatics, we believe that purification of RS3 complex and LRRC23 interactome study is required, which is one of our future directions. Given the limitation of our current data, we choose to stay conservative and not to suggest detailed structural information of LRRC23 in RS3 complex.